# Impact of different sources of precursors on an ozone pollution outbreak over Europe analysed with IASI+GOME2 multispectral satellite observations and model simulations

Sachiko Okamoto[1], Juan Cuesta[1], Matthias Beekmann[2], Gaëlle Dufour[2], Maxim Eremenko[1], Kazuyuki Miyazaki[3], Cathy Bonne[4], Hiroshi Tanimoto[5], Hajime Akimoto[5]

[1] Univ Paris Est Creteil and Université de Paris Cité, CNRS, LISA, F-94010 Créteil, France
[2] Université de Paris Cité and Univ Paris Est Creteil, CNRS, LISA, F-75013 Paris, France
[3] Jet Propulsion Laboratory (JPL), California Institute of Technology, Pasadena, 91109, CA, USA
[4] Institut Pierre Simon Laplace (IPSL), AERIS data centre, Paris, 75252, France
[5] National Institute for Environmental Studies, Tsukuba, 350-8506, Japan

*Correspondence to*: Sachiko Okamoto (sachiko.okamoto@lisa.ipsl.fr) and Juan Cuesta (cuesta@lisa.ipsl.fr)

**Abstract.** We examine the impact of different sources of ozone precursors on the daily evolution of successive ozone pollution outbreaks across Europe in July 2017 by using a multispectral satellite approach called IASI+GOME2, and a tropospheric chemistry reanalysis named TCR-2. IASI+GOME2, combining IASI (Infrared Atmospheric Sounding Interferometer) and GOME-2 (Global Ozone Monitoring Experiment-2) measurements respectively in the infrared and the ultraviolet, allows the observation of the daily horizontal distribution of ozone in the lowermost troposphere (defined here as the atmospheric layer between the surface and 3 km above sea level). IASI+GOME2 observations show a fair capacity to depict near-surface ozone evolution as compared to surface measurements from 188 European stations for the period 15–27 July 2017.

At the beginning of this event (on 16 July), an ozone outbreak is initially formed over the Iberian Peninsula likely linked with high temperature-induced enhancements of biogenic volatile organic compounds concentrations and collocated anthropogenic emissions. In the following days, the ozone plume splits into two branches, one being transported eastward across the Western Mediterranean and Italy, and the other one over Western and Central Europe. The southern branch encounters ozone precursors emitted over the Balkan Peninsula by wildfires along the coast of the Adriatic Sea and biogenic sources in the inland region of the Peninsula. Ozone concentrations of the northern plume enhance by photochemical production associated with anthropogenic sources of ozone precursors over Central Europe and by mixing with an ozone plume arriving from the North Sea that was originally produced over North America. Finally, both ozone branches are transported eastwards and mix gradually, as they reach the northern coast of the Black Sea. There, emissions from agricultural fires after harvesting clearly favor photochemical production of ozone within the pollution plume, which is advected eastwards in the following days. Based on satellite analysis, this paper shows the interplay of various ozone precursor sources to sustain a two-week long ozone pollution event over different parts of Europe.

# 1 Introduction

Tropospheric ozone is one of the key gases in the Earth's atmosphere because it plays a significant role in global warming (Szopa et al., 2021) and in determining the oxidizing capacity of the troposphere (Monks et al., 2015). Tropospheric ozone is also recognized as the gaseous air pollutant that causes the greatest threat to human and ecosystem health (Monks et al., 2015). The lifetime of ozone in the troposphere ranges from a few hours in polluted urban regions to up to few months in the upper troposphere, but it is in average relatively long (~22 days; Young et al., 2013). This allows tropospheric ozone transport over distances of intercontinental and hemispheric scales. The main source of tropospheric ozone is in situ photochemical production through oxidation of non-methane volatile organic compounds (NMVOCs), carbon monoxide (CO) and methane ($CH_4$), in the presence of nitrogen oxides ($NO_X$) (e.g., Atkinson, 2000). Stratosphere-to-troposphere transport of ozone also significantly enhances its abundance in the troposphere (e.g., Stohl et al., 2003). Photochemical formation of ozone occurs through a series of complex, nonlinear reactions involving sunlight, $NO_X$, volatile organic compounds (VOCs), and oxidation of free radical species (Chameides and Walker, 1973). Generally, local concentrations of either radicals or $NO_X$ are sufficiently high so that the other species are chemically limiting the formation of ozone. These two photochemical regimes are commonly referred to $NO_X$-limited and VOC-limited ($NO_X$-saturated or radical-limited) (Sillman et al., 1990; Kleiman, 1994). At regional and global scales, ozone production is largely $NO_X$-limited, although urban areas with high $NO_X$ emissions are frequently VOC-limited. However, the sensitivity of local ozone production for individual locations and events is often uncertain. Understanding the complex spatial and temporal evolution of photochemical regimes at local scales is still an important issue.

The magnitude of local air pollution is strongly driven by local and regional emissions of air pollutants and their precursors. The precursors of ozone are emitted from multiple sources associated with anthropogenic activities, biogenic origin and biomass fires. In Europe, regional anthropogenic emissions have been controlled by governmental regulations and dropped considerably between 1990 and 2018 (EEA, 2020). $NO_X$ emissions decreased (–60 %) in the electricity/energy generation sectors because of certain technical measures such as the introduction of combustion modification technology, the implementation of fuel gas abatement techniques, and fuel switching from coal to gas. NMVOCs emissions also decreased (–62 %) in the road transport sector. As a result of these reductions, the percentage of European Union (EU) urban population potentially exposed to high ozone concentrations (above the EU target value for protecting human health) have declined from 2002 to 2011 (Guerreiro et al., 2014). Additionally, summer peaks of ozone concentrations have also reduced over rural areas of Europe (~0.46 ppb yr$^{-1}$ from 1995 to 2014; Yan et al., 2018). However, background ozone concentrations (5th percentile) have increased continually over Europe during the last years (~0.15 ppb yr$^{-1}$; Yan et al., 2018; 2019). This may be partly linked with enhancements of ozone levels in urban areas owing to a reduction of the urban ozone sink by titration, which is associated with the reduction of $NO_X$ (Sicard et al., 2013; Yan et al., 2018; 2019). Moreover, the amount of ozone production not only responds nonlinearly to changes in precursor emissions but also it is sensitive to variations in air temperature, radiation, and other climate factors (e.g., Jacob and Winner, 2009). Especially, the impacts of temperature on ozone occur both directly and indirectly. The relationship between ozone and the precursors is driven by several well-known temperature-dependent

mechanisms: the thermal decomposition of the $NO_X$ reservoir species peroxyacetyl nitrate (PAN, $CH_3COO_2NO_2$) (Orlando et al., 1992), the temperature-dependent emissions of both VOCs from vegetation and $NO_X$ from soil (Vautard et al., 2005), the extra evaporation of anthropogenic VOCs at high temperature (Fehsenfeld et al., 1992; Simpson, 1995), or the high stomatal resistance in warm conditions, which limits the dry deposition of ozone on vegetation (Solberg et al., 2008).

Satellite observations offer a great potential for overcoming the limited spatial coverage of ground-based measurements and

70 filling the observational gap of air pollution. Nevertheless, measuring ozone pollution from space has been a challenge because standard single-band ozone retrievals cannot provide quantitative information of the ozone concentrations within the planetary boundary layer (PBL). Spaceborne spectrometers operating in the ultraviolet (UV), like GOME-2 (Global Ozone Monitoring Experiment-2), have been used to derive observations of tropospheric ozone with maximum sensitivity around 5–6 km altitude (e.g., Liu et al., 2010; Cai et al., 2012). Thermal infrared (TIR) spaceborne instruments, like IASI (Infrared Atmospheric

Sounding Interferometer) onboard the MetOp satellites, have shown good performance for observing ozone in the lower troposphere, but with sensitivity peaking at 3 km altitude at the lowest (e.g., Eremenko et al., 2008; Dufour et al., 2012). Recently, synergetic approaches simultaneously using UV and TIR radiances has been developed for enhancing the sensitivity to lower tropospheric ozone (e.g., Cuesta et al., 2013; Fu et al., 2013; 2018; Colombi et al., 2021). A multispectral approach called IASI+GOME2, combining IASI observations in the TIR and GOME-2 measurements in the UV, showed remarkable

skills for observing the horizontal distribution of ozone concentrations in the lowermost troposphere (LMT), hereafter defined as the atmospheric layer between the surface and 3 km above sea level (a.s.l.) (Cuesta et al., 2013; 2018). Cuesta et al. (2018) showed that IASI+GOME2 has air-quality-relevant skills to quantitatively describe the transport pathways, the daily evolution and photochemical production of lowermost tropospheric ozone during a major outbreak across East Asia.

In the present paper, we characterize an ozone outbreak across Europe in July 2017 by using the multispectral satellite approach

IASI+GOME2, tropospheric chemistry reanalyses and other observations. This is a moderate pollution event, which is very common during the summer in this region (e.g., Cuesta et al., 2013; Foret et al., 2014; Kalabokas et al., 2020). We analyse the sources that have most probably contributed to ozone photochemical production along transport of moderate ozone plumes travelling from Southwest to Eastern Europe. To the authors' knowledge, such detailed identification of the sources of ozone precursors of a complex event over Europe has not yet been performed using satellite observations. Section 2 describes the

observational and modelling data used for the analysis. Results and discussions on the evolution of the European ozone outbreak and the associated precursor sources are presented in section 3. A summary and conclusions are given in the last section.

## 2 Data and methods

### 2.1 Satellite observations of lowermost tropospheric ozone from IASI+GOME2

The multispectral satellite approach IASI+GOME2 was designed for observing lowermost tropospheric ozone by synergism of TIR atmospheric radiances observed by IASI and UV earth reflectances measured by GOME-2 (Cuesta et al., 2013; 2018). Both instruments are onboard the MetOp satellite series, and they both offer global coverage every day (for MetOp-A around 09:30 local time and for MetOp-B around 09:00 local time) with a relatively fine ground resolution (12 km diameter pixels spaced by 25 km for IASI at nadir and ground pixels of 80 km × 40 km for GOME-2). The IASI+GOME2/MetOp-B product

including vertical profiles of ozone is publicly available on the French data centre AERIS (https://iasi.aeris-data.fr, last access: 7 November 2022). We use an updated version of the IASI+GOME2 product, merging the IASI+GOME2/MetOp-A and IASI+GOME2/MetOp-B products for improving the spatial coverage. For reducing random errors, the dataset is averaged in a regular horizontal grid of 1° × 1°, in the same way as done by Cuesta et al. (2018). This spatial resolution enables the observation of the horizontal distribution of regional-scale ozone plumes, such as those analysed in the current study over

Europe. Clear sky conditions ensure good daily coverage of satellite data, as was the case during the analysed event. Ozone concentrations from the surface to 3 km of altitude (a.s.l.) are provided as an average ozone volume mixing ratio in ppb within the layer, which is calculated as the ratio of lowermost tropospheric partial columns (in molecules per square centimetre) of ozone and air. Hereinafter, this amount is designated as IASI+GOME2 LMT ozone concentration. In addition, we use IASI+GOME2 ozone observations in the upper troposphere from 6 to 12 km a.s.l. to examine ozone-rich air masses in the

middle and upper troposphere.

For assessing the contribution of IASI+GOME2 for capturing near surface ozone evolution, we compared IASI+GOME2 with in situ surface observations of ozone registered in the Tropospheric Ozone Assessment Report (TOAR) database (Schultz et al., 2017). TOAR is a research community with an up-to-date scientific assessment of tropospheric ozone's global distribution and trends from the surface to the tropopause initiated by the International Global Atmospheric Chemistry (IGAC). The data

in this database is provided by a few well-managed networks, and their metadata and qualities are examined by the TOAR data centre (Schultz et al., 2017). In the present study, we use hourly surface data from stations in rural areas, derived from in situ measurements performed using the UV absorption technique. As hourly ozone data are not currently distributed by the TOAR portal, we obtained this data from the following sources: the Global Atmosphere Watch (GAW, http://ebas.nilu.no, last access: 28 January 2020), the European Monitoring and Evaluation Programme (EMEP, http://ebas.nilu.no, last access: 28

January 2020), and the European Air Quality e-Reporting (https://eea.europa.eu, last access: 27 January 2020). We identify rural stations based on the TOAR station category (Schultz et al., 2017). The TOAR station category characterizes stations as "urban", "rural, low elevation" (located below 1500 m a.s.l.), "rural, high elevation" (above 1500 m a.s.l.) and "unclassified" based on several high-resolution global gridded data products. Here we only use data from "rural, low elevation" and "rural, high elevation" stations. We consider afternoon averages (12:00–16:00 local time - LT) of these surface concentrations that

are expected to be vertically mixed within the mixing atmospheric boundary layer. This is probably more comparable to the

IASI+GOME2 retrievals than morning surface concentrations that have not been mixed within the whole boundary layer (Cuesta et al., 2022). This is explained by the fact that the IASI+GOME2 LMT retrievals are sensitive around ~2.2 km of altitude during summer over land (see Cuesta et al., 2013), thus mostly measuring ozone concentrations at the residual atmospheric boundary layer. It is worth noting that as for other ozone satellite retrievals, the height of maximum sensitivity of

IASI+GOME2 vary depending on the observing conditions. Due to the use of thermal infrared measurements, the height of sensitivity over the ocean is 1 to 2 km higher than that over land (as the thermal contrast between the near-surface air and the surface itself is weaker over the ocean than over land).

The comparison of IASI+GOME2 data with respect to surface in situ measurements over Europe is presented in Figures 1–2 and Table 1, for the period 15–27 July 2017 (one day before and after the moderate ozone outbreak analysed in the paper).

Figure 1 shows the correlation coefficients $R$ between daily in situ surface observations at each individual station (squares or circles) and colocalized IASI+GOME2 retrievals. We consider 188 stations (18 high elevation and 170 low elevation stations). Generally, the correlation coefficients are positive, typically ranging from 0.3 to 0.8 and with approximately 54 % of the sites having statistically significant correlations (with $P$-values of significance test < 0.05). Relatively higher correlation coefficients are found in the Western Mediterranean Basin, Central France, and Austria, whereas they are relatively lower over Western

Iberian Peninsula, North-western France and the Central Mediterranean Basin.

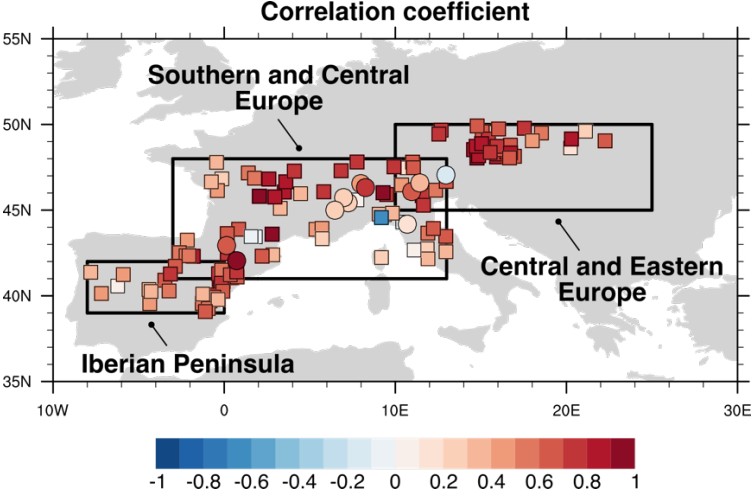

**Figure 1:** Correlation coefficients between surface ozone measurements at each individual station and colocalized IASI+GOME2 retrievals for the period 15–27 July 2017. Circles and squares indicate respectively high (above 1500 m a.s.l.)
and low elevation stations based on the TOAR category. Black rectangles indicate the subregions considered in Figure 2 and Table 1 for comparisons between IASI+GOME2 and surface data.

Figure 2 shows the scatter plots of daily in situ surface observations from the ensemble of stations within three sub-regions (shown as black rectangles in Fig. 1) with IASI+GOME2 data colocalized in time (at daily scale) and in space (satellite data at 1° × 1° sampled at the location of each station) for the period 15–27 July 2017. These sub-regions are (i) the Iberian Peninsula (39°–42°N, 0°–8°W), (ii) Southern and Central Europe (41°–48°N, 3°W–13°E), and (iii) Central and Eastern Europe (45°–50°N, 10°–25°E). The statistics for each of the three sub-regions are shown in Table 1. All comparisons show significant correlations (*P*-value < 0.01). IASI+GOME2 over the Iberian Peninsula shows a small positive mean difference of 0.9 ppb with respect to surface data. Over Southern and Central Europe and Central and Eastern Europe, the satellite retrievals depict smaller values than in situ measurements by –4.7 ppb and –10.8 ppb on average, respectively. The root mean squared (RMS) difference over the Iberian Peninsula is smaller than those in the other two regions. The values of correlation coefficients around ~0.45 are not very high, but they may reflect differences in the spatio-temporal representativity of surface in situ measurements at each station and colocalized satellite data at daily scale for this particular event. Vertical gradients of ozone concentrations near the surface may also be responsible of the apparent differences between satellite and surface data. Lower ozone abundances retrieved by the satellite approach may be induced by high ozone concentrations near the surface, as typically remarked during ozone outbreaks of anthropogenic origin. Previous comparisons of IASI+GOME2 data with surface in situ measurements showed relatively larger values of correlation coefficients, being ~0.69 for a major ozone outbreak over Eastern Asia (11 surface stations over Japan, Cuesta et al., 2018) and ~0.55 for differences of 15-day averages over Europe (Cuesta et al., 2022). In this last case, ozone satellite retrievals over Europe showed lower concentrations than at the surface (by –8.6 ppb over the whole continent), as found for the event analysed here.

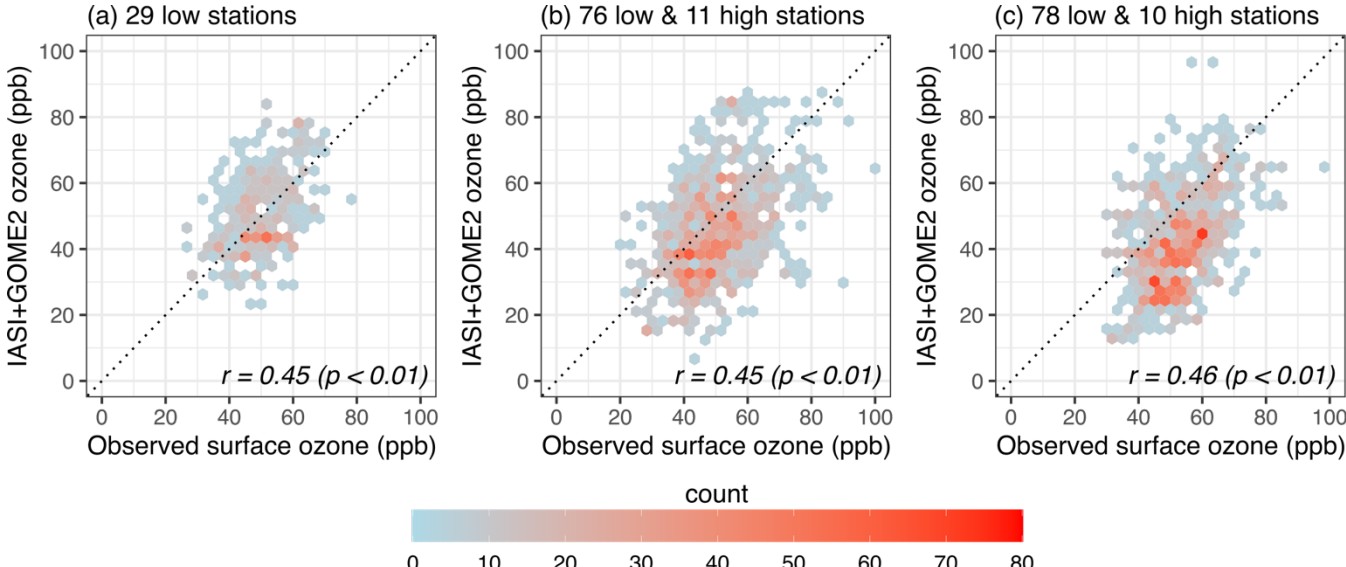

**Figure 2:** Comparison of IASI+GOME2 with surface observations of ozone for the period 15–27 July 2017. Colocalization of the two datasets is done at daily scale and for the location of individual surface stations, while scatter plots show all

coincident data for the ensemble of stations within the following sub-regions (indicated as black rectangles in Fig. 1): **(a)** the Iberian Peninsula, **(b)** Southern and Central Europe, and **(c)** Central and Eastern Europe. The colour scale represents the number of occurrences of points within each bin (of $3.3 \times 3.3$ ppb).

**Table 1:** Statistics of comparison of in situ surface observations with respect to IASI+GOME2 ozone in three sub-regions for the period 15–27 July 2017**.** Colocalization of the two datasets is done at daily scale and for the location of individual surface stations, while the values in the table are calculated for all coincident data for the ensemble of stations within the following sub-regions (indicated as black rectangles in Fig. 1).

|  | Iberian Peninsula | Southern and Central Europe | Central and Eastern Europe |
|---|---|---|---|
| Mean difference (ppb) | 0.9 | –4.7 | –10.8 |
| RMS difference (ppb) | 11.8 | 15.1 | 16.9 |
| R | 0.45 (*P*-value < 0.01) | 0.45 (*P*-value < 0.01) | 0.46 (*P*-value < 0.01) |

## 2.2 Tropospheric Chemistry Reanalysis version 2 (TCR-2)

Chemical reanalyses provide comprehensive information on the evolution of atmospheric composition, offering the spatio-temporal full coverage of chemistry-transport models and enhanced precision by assimilation of various satellite observations. Data assimilation provides an estimate of the most likely state of system (Lahoz and Schneider, 2014). As support of the present analysis of the link between ozone precursors and an ozone outbreak over Europe, we use the Tropospheric Chemical Reanalysis version 2 (TCR-2). This is a tropospheric chemical reanalysis based on an assimilation of multi-constituent observations from diverse satellite sensors (Miyazaki et al., 2020a). TCR-2 data is publicly available for the period 2005–2019 (https://tes.jpl.nasa.gov/tes/chemical-reanalysis/). In the present study, we use 2-hourly TCR-2 reanalysis data (Miyazaki, personal communication, 2020) of ozone and two species representative of ozone precursors: nitrogen dioxide ($NO_2$) and formaldehyde (HCHO) as proxy for $NO_X$ and VOCs distributions, which provide supplemental information on the evolution and origin of anthropogenic air pollution. TCR-2 has already been employed in various air quality studies (e.g., Kanaya et al., 2019; Miyazaki et al., 2019; 2020b).

TCR-2 uses MIROC-CHASER (Model for Interdisciplinary Research on Climate-Chemical atmospheric general circulation model for study of atmospheric environment and radiative forcing, Watanabe et al., 2011) as a base forecast model, which contains detailed photochemistry in the troposphere and stratosphere by simulating tracer transport, wet and dry depositions, and emissions. The model calculates the concentrations of 92 chemical species and 262 chemical reactions (58 photolytic, 183 kinetic, and 21 heterogeneous reactions). Its tropospheric chemistry considers the fundamental chemical cycle of $O_X$-$NO_X$-$HO_X$-$CH_4$-CO along with oxidation of NMVOCs to properly represent ozone chemistry in the troposphere. TCR-2 has a T106

horizontal resolution (1.1° × 1.1°) with 32 vertical levels from the surface to 4.4 hPa. Nine of these vertical levels are typically found within the lowermost troposphere (below 3 km of altitude). In this study, we mainly use model data at 850 hPa for describing the abundance of ozone precursors and the pathways of ozone plumes at this atmospheric level. Simulations from 1000 hPa to 200 hPa also allow the investigation of the influence of downward transport from the upper troposphere and the lower stratosphere. The model-derived ozone and CO data is used here without smoothing by averaging kernels of the satellite retrievals (as also done by e.g., Foret et al., 2014; Cuesta et al., 2018). This model data provides information with fine vertical resolution on the spatial distribution at these atmospheric pollutants at specific vertical levels. Therefore, they are complementary with respect to satellite retrievals, that represent the abundances of these pollutants integrated vertically within an atmospheric layer (i.e., the LMT for ozone and the total column of CO, described respectively in subsections 2.1 and 2.2) and their sensitivity may vertically vary depending on the observing conditions. The combined use of model and satellite data allow accounting for the possible uncertainties or limitations of each of them, while providing a more complete description of the spatio-temporal evolution of ozone precursors.

Meteorological fields used by TCR-2 are nudged towards the 6-hourly ERA-Interim (Dee et al., 2011). A priori emissions of $NO_X$, CO, and $SO_2$ are obtained from bottom-up emission inventories. Anthropogenic $NO_X$, CO and $SO_2$ emissions are obtained from the Hemispheric Transport of Air Pollution (HTAP) v2 for 2010 (Janssens-Maenhout, et al., 2015). For biomass burning emissions, the monthly Global Fire Emissions Database (GFED) v4 (Randerson et al., 2018) are used. Emissions from soils are based on monthly mean Global Emissions Inventory Activity (GEIA) (Graedel et al., 1993). Data assimilation is based on an ensemble Kalman filter (EnKF) approach, the Local Ensemble Transform Kalman Filter (LETKF) (Hunt et al., 2007) for ozone ($O_3$), CO, $NO_2$, $HNO_3$ and $SO_2$. Tropospheric $NO_2$ column retrievals used for data assimilation are the QA4ECV version 1.1 level (L2) product for the Ozone Monitoring Instrument (OMI) (Boersma et al., 2017a), GOME-2 (Boersma et al., 2017b), and the Scanning Imaging Absorption Spectrometer for Atmospheric Cartography (SCIAMACHY) (Boersma et al., 2017c). Ozone retrievals are taken from the version 6 level 2 nadir data obtained from the Tropospheric Emission Spectrometer (TES) (Bowman et al., 2006; Herman and Kulawik, 2020) and the version 4.2 for the Microwave Limb Sounder (MLS) for pressures of lower than 215 hPa (Livesey et al., 2011, 2020). Total column CO data are the version 7 L2 TIR/NIR product for the Measurements of Pollution in the Troposphere (MOPITT) (Deeter et al., 2017). The performance of the chemical reanalysis products has been validated against various independent surface and aircraft measurements (Miyazaki et al., 2020a).

### 2.3 Other satellite observations and emission inventories

For analysing the origin and evolution of ozone plumes, CO retrievals from IASI/MetOp-B are used for identifying the location of CO plumes. This total column CO product is publicly available at the French data centre AERIS (https://iasi.aeris-data.fr, last access: 7 November 2022). It is derived from IASI radiance using the FORLI algorithm (Fast Optimal Retrievals on Layer for IASI; Hurtmans et al., 2012), from the Université Libre de Bruxelles (ULB) and the Laboratoire Atmosphères, Milieux,

Observations Spatiales (LATMOS). Cuesta et al. (2018) used the same product together with IASI+GOME2. For reducing random errors, the dataset was averaged in regular grid of $1° \times 1°$ in the same way as IASI+GOME2. For consistency with the other datasets, we derive CO mixing ratios assuming that all CO molecules within the column are located in the lowermost troposphere. Overestimations may occur for CO plumes extending above the LMT and because of the presence of background CO in the free troposphere. This is not expected to impact our results, as these observations are mainly qualitatively used.

We analyse the sources of ozone precursors using anthropogenic and biogenic emissions from the Emissions of Atmospheric Compounds and Compilation of Ancillary Data (ECCAD, https://eccad.aeris-data.fr/, last access: 26 May 2021). Anthropogenic $NO_X$ and NMVOCs emissions were obtained from CAMS Global anthropogenic emissions (CAMS-GLOB-ANT v4.2-S1.1; Granier et al., 2019). The emissions are based on the emissions provided by the Emissions Database for Global Atmospheric Research (EDGAR v4.3.2; Crippa et al., 2018) and the Community Emissions Data System (CEDS; Hoestly et al., 2018), and provided as monthly averages with a horizontal resolution of $0.1° \times 0.1°$. Soil NO emission was obtained from CAMS Global soil emissions provided as monthly averages with a horizontal resolution of $0.5° \times 0.5°$ (CAMS-GLOB-SOIL v2.3; Granier et al., 2019; Simpson and Darras, 2021). Soil $NO_X$ is mainly emitted as NO. The Model of Emissions of Gases and Aerosols from Nature (MEGAN) is one of the most widely used biogenic emission model (Guenther et al., 2012). We obtained biogenic isoprene and monoterpenes emissions developed under the Monitoring Atmospheric Composition and Climate project (MACC), MEGAN-MACC Biogenic emission inventory (MACC-MEGAN, Sindelarova et al., 2014). The emissions are provided as monthly averages with a horizontal resolution of $0.5° \times 0.5°$.

The locations of fires are derived from the Terra and Aqua MODIS (Moderate Resolution Imaging Spectrometer) active fire products (MCD14ML Collection 6; Giglio et al., 2016) distributed by the Fire Information for Resources Management System (FIRMS, https://earthdata.nasa.gov/firms, last access: 13 March 2020). This dataset provides the values of Fire Radiative Power (FRP) and the inferred hot spot type: "presumed vegetation fire", "active volcano", "other static land source", and "offshore". We only used the FRP values of "presumed vegetation fire". In addition to the active fire product, we examine the type of fire with the Terra and Aqua combined MODIS Land Cover Climate Modeling Grid (CMG) (MCD12C1) Version 6 data product (Friedl and Sulla-Menashe, 2015). The global land cover is distributed by the Land Processes Distributed Active Archive Center (LP DAAC, https://lpdaac.usgs.gov/, last access: 25 May 2021) at yearly intervals with horizontal resolution of $0.05° \times 0.05°$. We adopt the International Geosphere-Biosphere Programme (IGBP) classification scheme.

**2.4 Meteorological and trajectory analysis**

Meteorological conditions leading to photochemical production of ozone and transport are described with ERA5 reanalysis (Hersbach et al., 2020) by the European Centre for Medium-Range Weather Forecast (ECMWF). These meteorological fields have global coverage, a horizontal resolution of $0.25° \times 0.25°$, 37 pressure levels and a time step of 1 hour (download from the Meso-centre IPSL through https://climserv.ipsl.polytechnique.fr, last access: 23 February 2021) and calculate several indices. Heatwaves are detected with a method adapted from Lavaysse et al. (2018), which is implemented in the Copernicus

European Drought Observatory (EDO). This indicator is calculated from daily maximum 2 m temperature ($T_{max}$). The threshold values of $T_{max}$ that characterize a heatwave are calculated from the observed $T_{max}$ for the calendar day during the extended summer (April–October) for a 30-year baseline period (1981–2010). The daily threshold values for $T_{max}$ are defined as the 90th percentile of 330 respective temperature values in an 11-day window centred on that day, for all years in the baseline period.

A heatwave is detected when there are at least three consecutive days with $T_{max}$ above its daily threshold value. When two successive heatwaves are separated in time by one day, they are considered as a single event and merged.

A classification of air masses originated at low latitude subtropical desertic areas is performed according to a method proposed by Sousa et al., (2019). The two following criteria identify the Saharan warm air intrusion: i) 1000–500 hPa layer geopotential thickness larger than 5800 m, and ii) 925–700 hPa layer potential temperature ($\theta$) greater than 40 °C. Grid points satisfying

both criteria correspond to low density, warm, stable, and very dry air masses, with the potential to be additionally warmed by downward advection. For confirming the occurrence of Saharan air intrusions, we analyse measurements from the Aerosol Robotic Network (AERONET) project, which is a ground-based aerosol remote sensing network (Holben et al., 1998). The aerosol optical depth (AOD) level 2 data from the Version 3 product is publicly available (https://aeronet.gsfc.nasa.gov, last access: 14 September 2020). The spectral variation of the AOD is expressed as the Angström exponent. This variable is known

as a qualitative indicator of aerosol particle size (Ångström, 1929); as small values (typically smaller than 0.5) indicate a greater abundance of coarse aerosols, such as desert dust and sea salt, and larger values suggest the presence of fine aerosols, typically from urban and biomass burning origins (e.g., Eck et al., 1999). This indicator is examined with measurements from Madrid (40.452°N, 3.724°W) and Granada (37.164°N, 3.605°W).

Air stagnation is characterized by stable weather conditions and weak winds in the lower to mid-troposphere and absence of

280 precipitation. It is often defined by three meteorological variables: upper-air speed, near-surface wind speed and precipitation. According to the previous method (Horton et al., 2012; Garrido-Perez et al., 2018; 2019), we considered a day and location as stagnant when the three following criteria are fulfilled simultaneously: daily mean windspeeds i) lower than 3.2 m s$^{-1}$ at 10 m and ii) lower than 13.0 m s$^{-1}$ at 500 hPa, and iii) daily total precipitation less than 1.0 mm.

Pathways of polluted air masses transported across Europe are estimated by the National Oceanic and Atmospheric

Administration (NOAA) Hybrid Single Particle Lagrangian Integrated Trajectory (HYSPLIT) model (Draxler and Hess, 1997; 1998; Draxler, 1999; Stein et al., 2015), driven by Global Data Assimilation System (GDAS) meteorological simulations (1° × 1° grid, from http://ready.arl.noaa.gov/archives.php, last access: 28 September 2020). We set the start height of calculation to 3000 m a.s.l. (which is the top of the LMT).

## 3 Results and discussion

In this section, we describe the ozone outbreak travelling across Europe during the period 15-27 July 2017. It is formed by three ozone plumes originating from the Iberian Peninsula, Western Europe and North America, that we call hereafter

according to their pathways across Europe respectively "southernmost", "middle latitude" and "northernmost" plumes. In a first subsection 3.1, we describe the transport patterns followed by these three ozone plumes over Europe. Then, we analyse the temporal evolution of the concentration of near-surface ozone and the link with those of three key precursors of ozone (subsection 3.2). This is followed by a detailed analysis of four key stages of the ozone outbreak: (subsection 3.3) the ozone outbreak over the Iberian Peninsula, (subsection 3.4) the impact of anthropogenic emissions in Western and Central Europe, (subsection 3.5) wildfires and biogenic emissions in the Balkan Peninsula and (subsection 3.6) agricultural burning emissions in the north coast of the Black Sea.

## 3.1 Pathways of the ozone plumes travelling across Europe

The pathways of these ozone plumes across Europe and the abundance of ozone precursors over this region is described in the following by a suite of daily maps of satellite observations (ozone at the LMT and total columns of CO) and TCR-2 simulations (at 850 hPa at 09:00 GMT) and supported by trajectories of the air mass associated with the pollutant plumes calculated by HYSPLIT for describing the trajectories of the plumes. These maps are shown in Figs. 3, 5, 6 and 7 for selected dates describing the main events during the evolution of the plumes.

The beginning of the ozone pollution outbreak is described in Figure 3 with daily maps on 16 July 2017 of ozone (Figs. 3a–b) and several key precursors (CO, $NO_2$ and HCHO respectively in Figs. 3c–e) and back-trajectories arriving on this day to the Iberian Peninsula (Fig. 3f). Observations from IASI+GOME2 depict an enhancement of ozone concentrations over the Iberian Peninsula during this day (indicated as a red rectangle in Fig. 3a). The location of the plume is defined according to the horizontal distributions of higher concentrations of ozone (including values roughly greater than ~60 ppb) than the surrounding background conditions, as depicted by IASI+GOME2. The travelling pathway of this ozone plume (and also for other plumes) is described by three-day forward trajectories calculated by HYSPLIT starting at the location of the observed ozone plumes every three days. According to these trajectories, the southernmost plume splits into two branches in the following days, one being transported eastward across the Western Mediterranean and Italy, and the other one over Western and Central Europe (see forward trajectories in Fig. 4e and red rectangle in 5a). In addition to this plume, two supplementary ozone plumes, called "middle latitude" and "northernmost" plumes, are observed over Western and Central Europe on 17–18 July (respectively black and blue rectangles in Fig. 5a). These two plumes are transported eastward across Western and Central Europe (Figs. 4a, c). On 20–22 July, the southern branch of the southernmost plume is transported eastward across the North-western Mediterranean (Fig. 4f and red rectangle in 6a). All ozone plumes move eastward and mix into a single large plume, and then arrive over the northern coast of the Black Sea on 23 July (Fig. 4b, d, g and red rectangle in 6a).

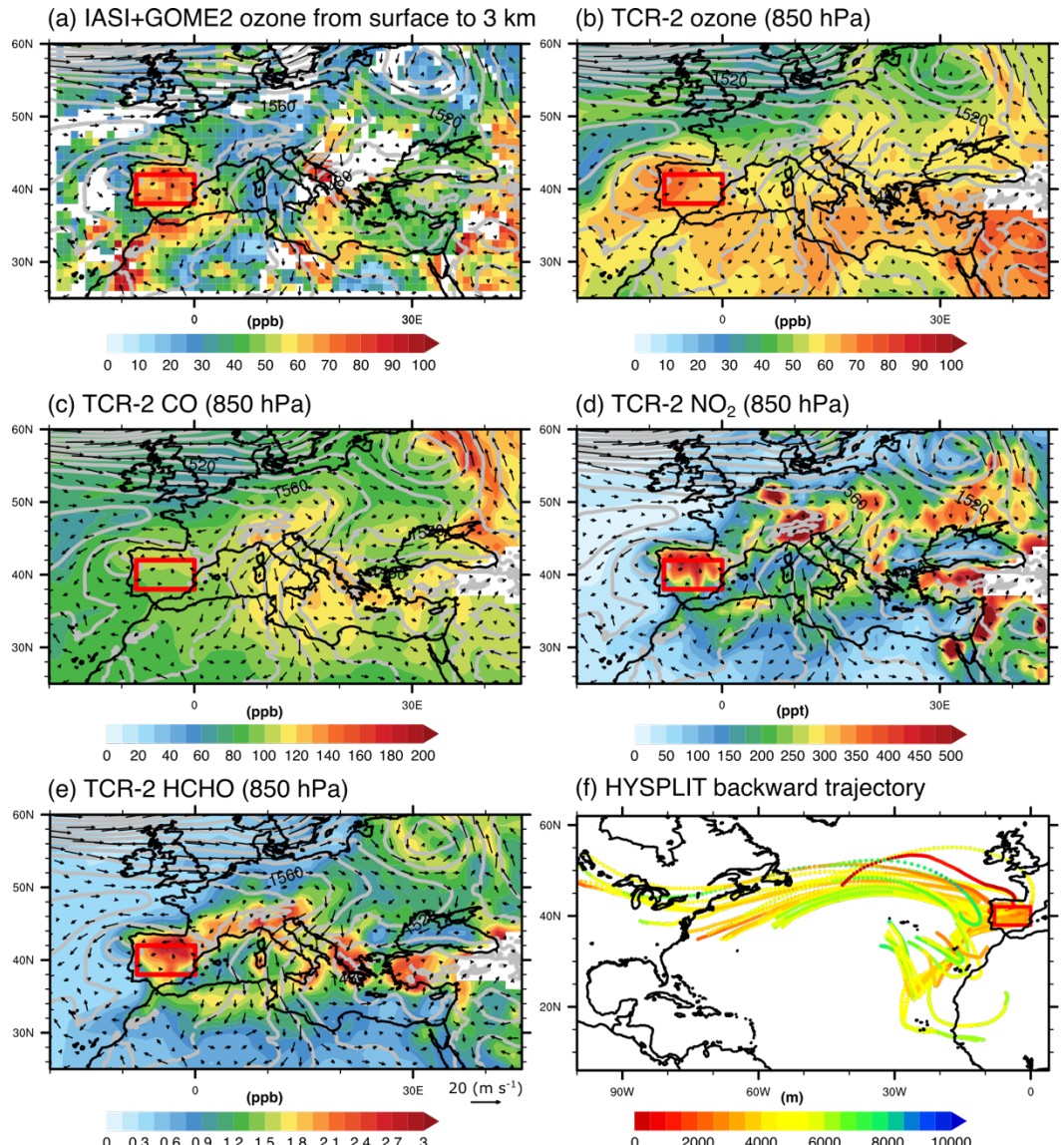

**Figure 3:** Distribution of gaseous pollutants and the meteorological conditions over Europe and the surrounding areas on 16 July 2017: **(a)** lowermost tropospheric ozone from IASI+GOME2 and **(b)** ozone, **(c)** CO, **(d)** NO₂, **(e)** HCHO from TCR-2 in the morning (at 09:00 GMT) at 850hPa, and **(f)** 7-day back trajectories arriving at 3000 m (a.s.l.) in the Iberian Peninsula. Winds and geopotential height at 850 hPa from ERA5 are indicated by black arrows and grey contour lines in panels (a–e). The red rectangles in (a–e) show the area for averaging concentrations of various air pollutants in Figure 8.

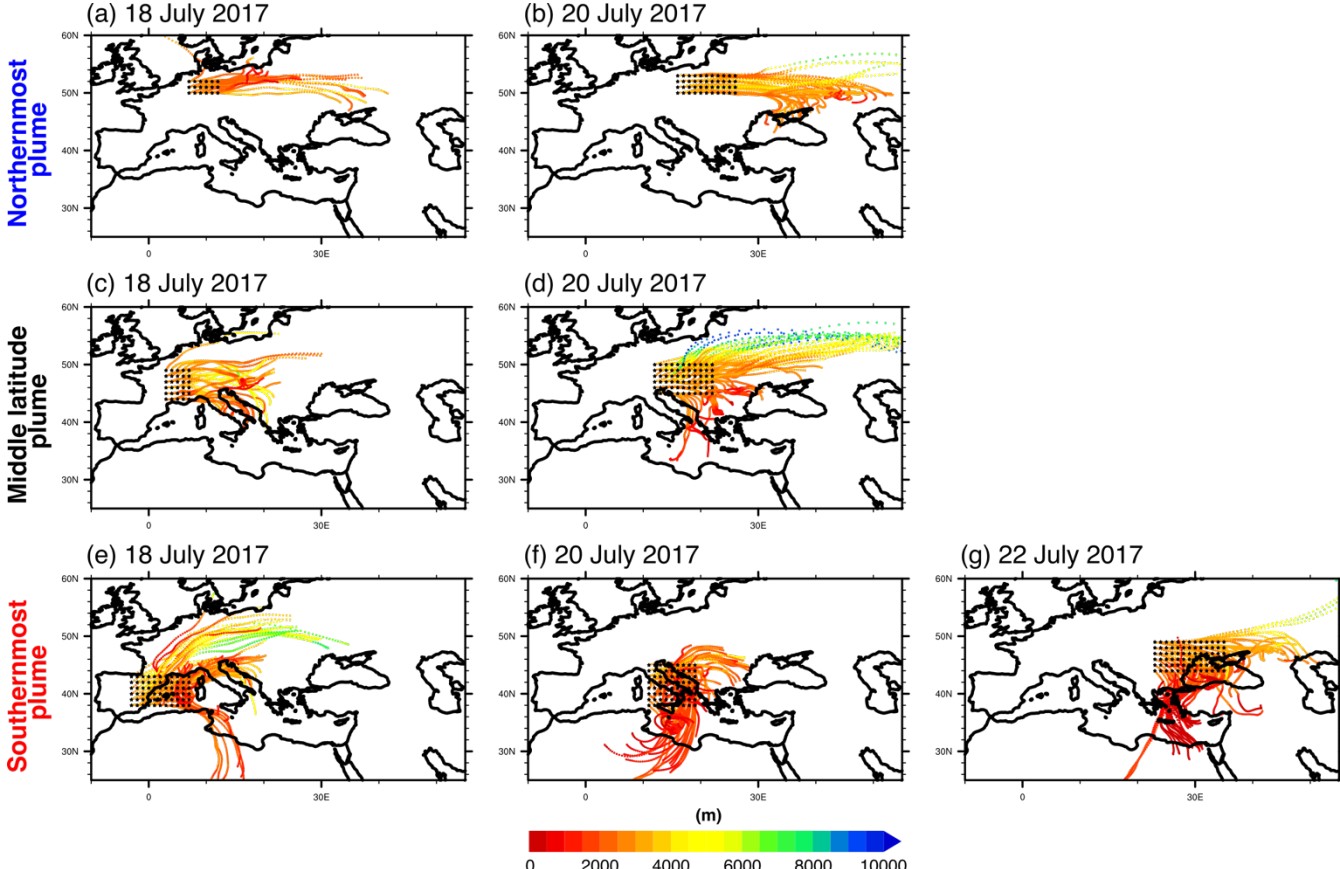

**Figure 4:** Three-day HYSPLIT forward trajectories depicting the pathways of the ozone plumes during the moderate ozone
outbreak in July 2017 over Europe. The tracks followed by the northernmost, middle latitude, and southernmost ozone plumes
are shown in the **(a and b)** upper, **(c and d)** middle and **(e, f and g)** lower panels. The trajectories start at 09:00 GMT on **(a, c and e)** 18, **(b, d and f)** 20 and **(g)** 22 July 2017, initiated at 3000 m (a.s.l.). Black points indicate the starting locations
corresponding the locations of ozone plumes.

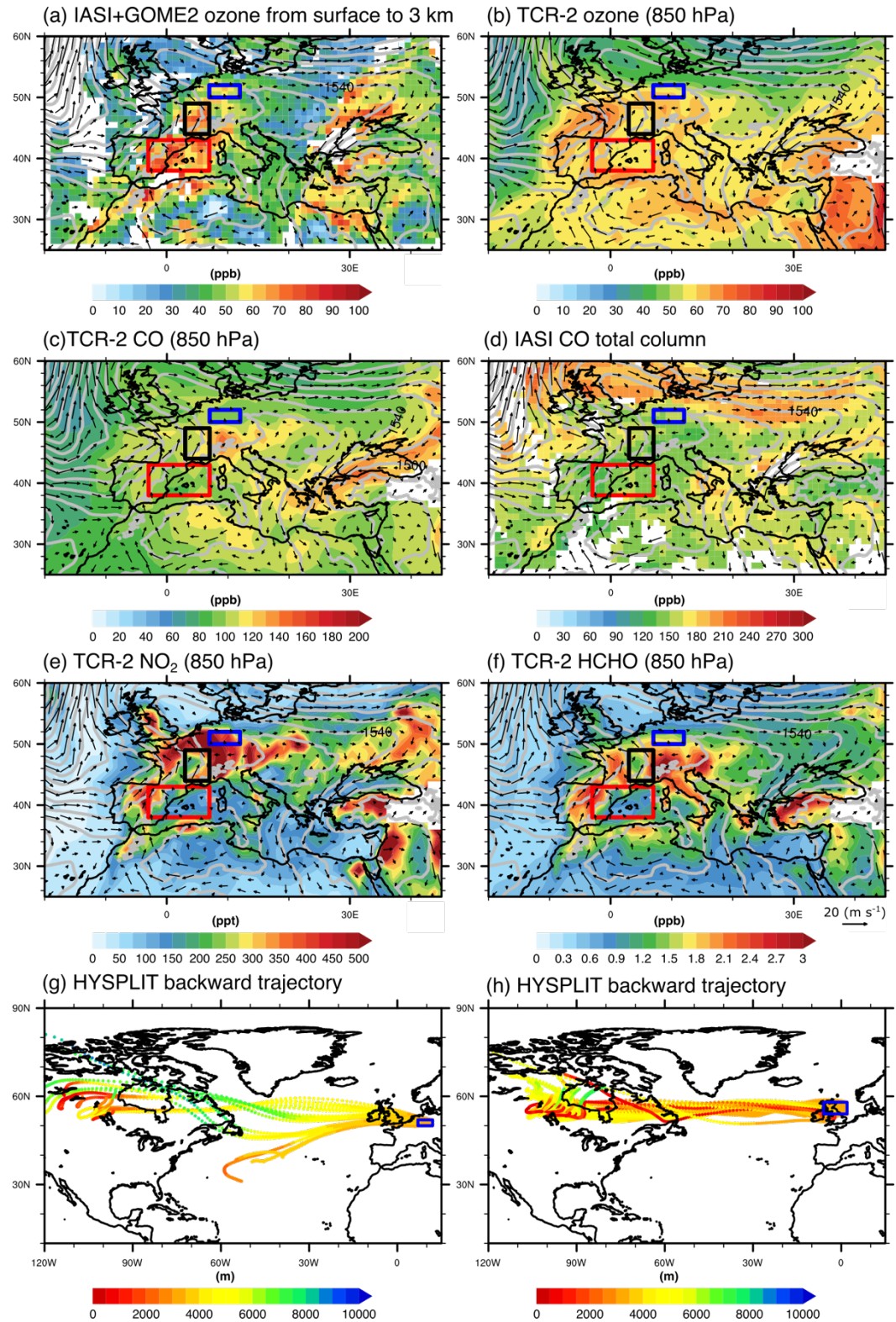

(a) IASI+GOME2 ozone from surface to 3 km  (b) TCR-2 ozone (850 hPa)

(c)TCR-2 CO (850 hPa)  (d) IASI CO total column

(e) TCR-2 NO$_2$ (850 hPa)  (f) TCR-2 HCHO (850 hPa)

(g) HYSPLIT backward trajectory  (h) HYSPLIT backward trajectory

**Figure 5:** Distribution of gaseous pollutants and the meteorological situation over Europe and the surrounding areas on 18 July 2017: **(a)** lowermost tropospheric ozone from IASI+GOME2, **(b)** ozone, **(c)** CO, **(e)** $NO_2$, and **(f)** HCHO from TCR-2 at 850hPa and **(d)** CO from IASI in the morning (at 09:00 GMT). Seven-day back trajectories arriving at 3000 m a.s.l. at the locations where ozone **(g)** and CO **(h)** plumes are observed.

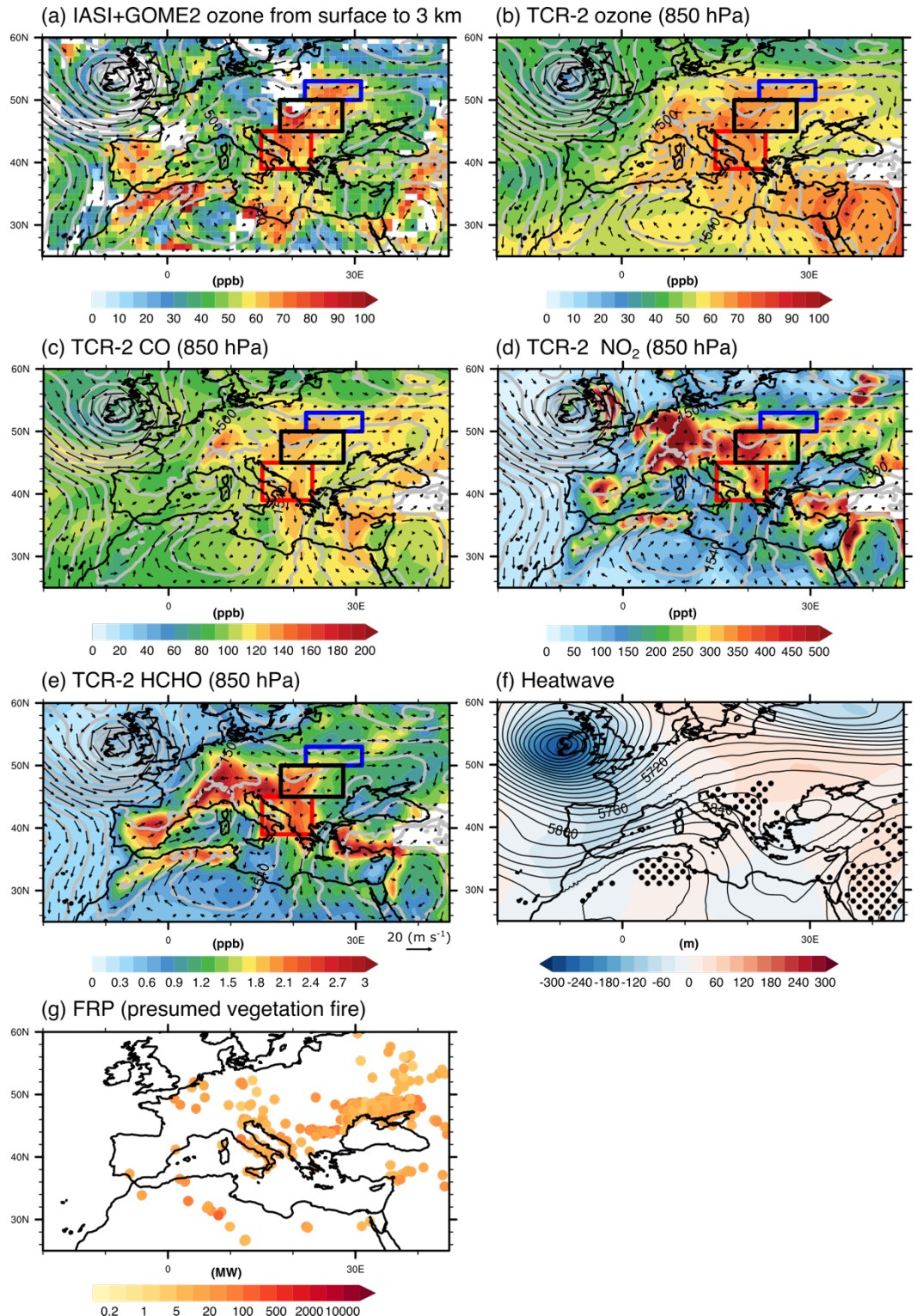

(a) IASI+GOME2 ozone from surface to 3 km
(b) TCR-2 ozone (850 hPa)
(c) TCR-2 CO (850 hPa)
(d) TCR-2 NO$_2$ (850 hPa)
(e) TCR-2 HCHO (850 hPa)
(f) Heatwave
(g) FRP (presumed vegetation fire)

**Figure 6: (a–e)** Same as Figure 3 but on 21 July 2017. **(f)** Heatwave indicators (black dots) and **(g)** fire radiative power (FRP) of presumed vegetation fire on 21 July 2017. Black contour lines and colour shades in (f) indicate daily mean and daily anomaly of geopotential height at 500 hPa, respectively.

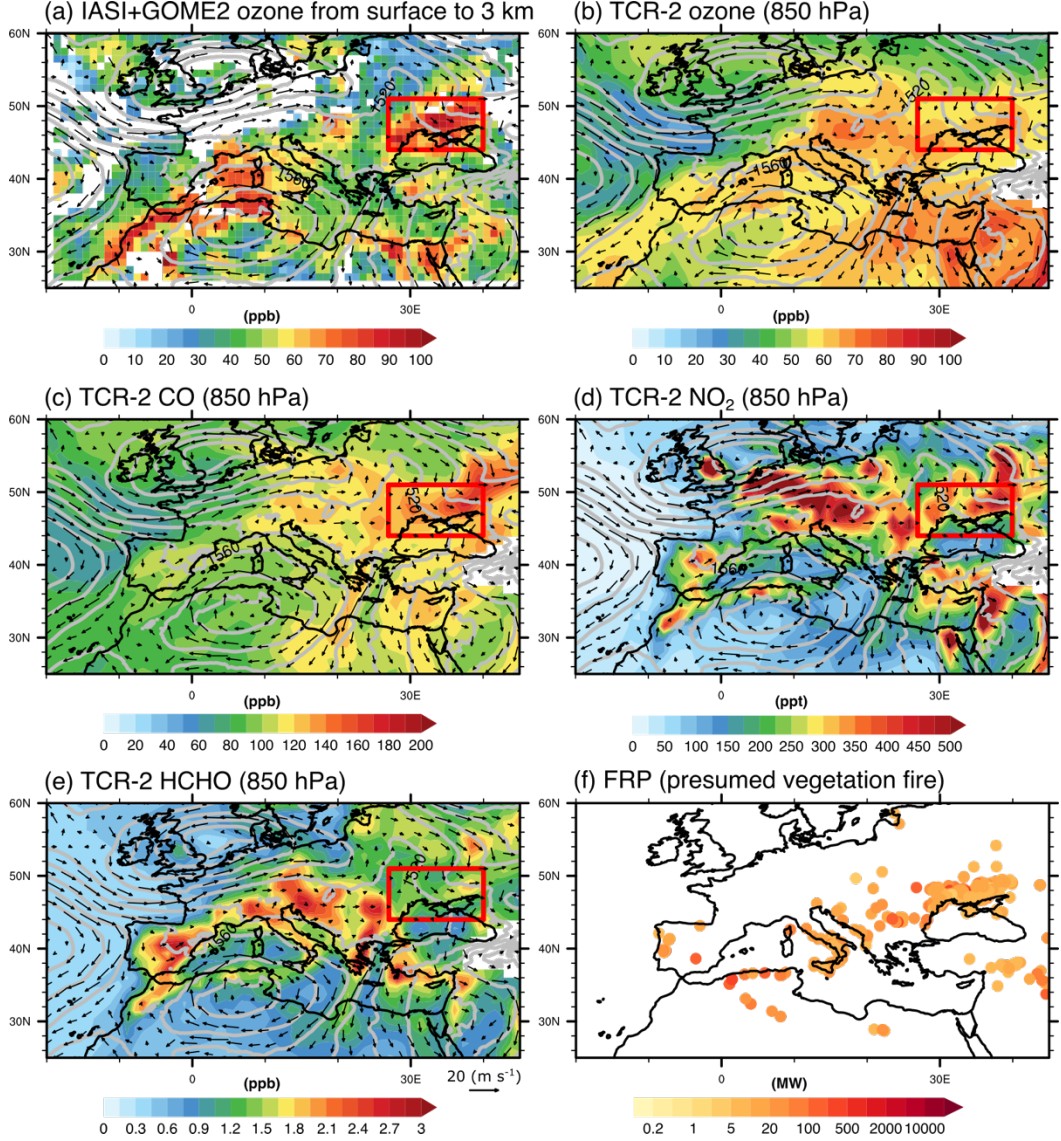

**Figure 7: (a–e)** Same as Figure 3a–e and **(f)** Same as Figure 6g but on 23 July 2017.

### 3.2 Temporal evolution of the air pollutant concentrations along the outbreak

Figure 8 shows a quantitative analysis of the Lagrangian evolution of the three ozone plumes, which can be clearly observed in IASI+GOME2 LMT ozone distributions for the period from 16 to 26 July 2017 (southernmost, middle latitude and northernmost plumes depicted by respectively red, black, and blue rectangles in Figs. 3, 5–7). These timeseries show the daily evolution of a given variable averaged at the location of highlighted ozone plumes (further discussions about their origin and evolution are provided in subsections 3.3 to 3.6). The ozone concentration in part of the plumes is often around 80 ppb, which

is near the ozone information threshold (whereby a 1-hour average concentration of 90 ppb — for a pressure of 1013 hPa and a temperature of 20 °C — triggers an obligation to inform the population on possible risk; EC, 2008). The ozone concentrations of these plumes are clearly enhanced by four dominant sources of ozone precursors, identified in this study. The southernmost plume is formed on 16 July and lasts until the end of the events on 26 July (red lines in Fig. 8). The initial formation of this moderately dense ozone plume is likely associated with enhanced biogenic emissions during the first two days, as depicted by

high concentrations of HCHO (red line in Fig. 8d). During this period, the ozone plume is co-located with low concentrations of CO, suggesting a limited influence of air pollutants emitted by combustion. Subsection 3.3 shows that this occurs over the Iberian Peninsula, where high surface temperature conditions prevail. Their increasing ozone concentrations are co-located with those of high CO content which clearly indicate a combustion-related origin of ozone precursors. As shown in subsection 3.4, the sources of these precursors are related to local anthropogenic activities in Southern and Central Europe (black lines in

Fig. 8; subsubsection 3.4.1) and transport of aged air masses from North America (blue lines in Fig. 8; subsubsection 3.4.2). On 21–22 July, the southern branch of the southernmost plume from the Iberian Peninsula is transported eastward across the North-western Mediterranean, exhibiting low concentrations of CO, $NO_2$ and HCHO (red line in Fig. 8c–e). The concentrations of these precursors increase as the plume approaches Italy and remain high while reaching the Balkan Peninsula. As shown in subsection 3.5, these last ones originate from wildfire emissions along the coast of the Adriatic Sea and biogenic emissions in

the inland region of the Balkan Peninsula. On 23 July, all ozone plumes mix into a single large plume, while they encounter very high CO concentrations (red line in Fig. 8c). These CO emissions are associated with active agricultural burning over the north coast of the Black Sea, as described in subsection 3.6.

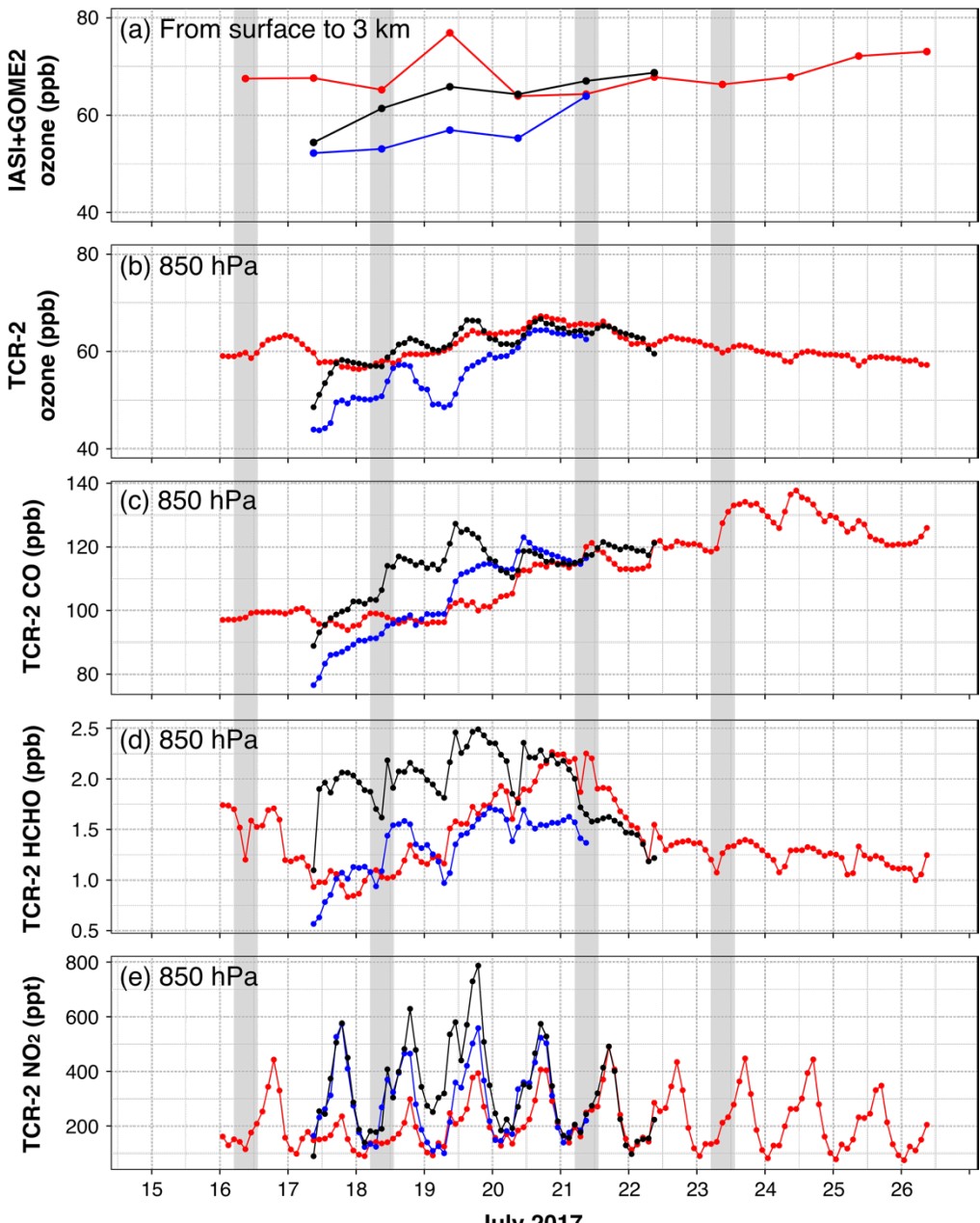

**Figure 8:** Lagrangian evolution of three air pollution plumes across Europe (red, blue and black rectangles in Figs. 3, 5–7) for the period from 16 to 26 July 2017 for the following variables:**(a)** lowermost tropospheric ozone from IASI+GOME2 and from TCR-2 at 850 hPa **(b)** ozone, **(c)** CO, **(d)** HCHO, and **(e)** $NO_2$. Red, blue, and black lines indicate the southernmost plume from the Iberian Peninsula, the northernmost plume from North America and the middle latitude plume from Western Europe, whose locations are depicted from the horizontal distribution of LMT ozone derived from IASI+GOME2. Gray shades indicate the date shown in Figures 3, 5–7.

### 3.3 Ozone outbreak over the Iberian Peninsula

### 3.3.1 Meteorological conditions

Figures 9 and 10 show the meteorological conditions which led to the ozone outbreak in July 2017. As typical for Iberian summer, an elevated layer of warm, dry and dusty air called Saharan air layer (SAL) is transported over the tropical Atlantic during the month of July 2017 (shown in Fig. S1 for 10 July 2017). The SAL is formed as warm and dry Saharan air moves westwards off the African coast, which is undercut by oceanic cooler and moister air of the marine boundary layer (see Fig. 9a–c, in similar conditions as described by e.g., Karyampudi and Carlson, 1988). While desertic air masses typically remain over the Sahara and the Atlantic, distant from Europe, an anticyclone occurring in mid-July 2017 favours the northward advection of these warm air masses. On 12 July, the high-pressure system is observed in terms of positive geopotential height anomalies at 500 hPa over the Iberian Peninsula, Northern Africa and North Atlantic (see isolines of geopotential height in Fig. 9d). On 13 and 14 July, the Saharan air intrusion reaches the Iberian Peninsula, as north as 40°N. The arrival of these desertic air masses is confirmed by sun photometer observations depicting large amounts of coarse particles, likely Saharan desert dust, over Central and Southern Iberian Peninsula (respectively at Madrid and Granada, Fig. 10c). This is suggested by low values of Ångström exponents measured on 11–19 July.

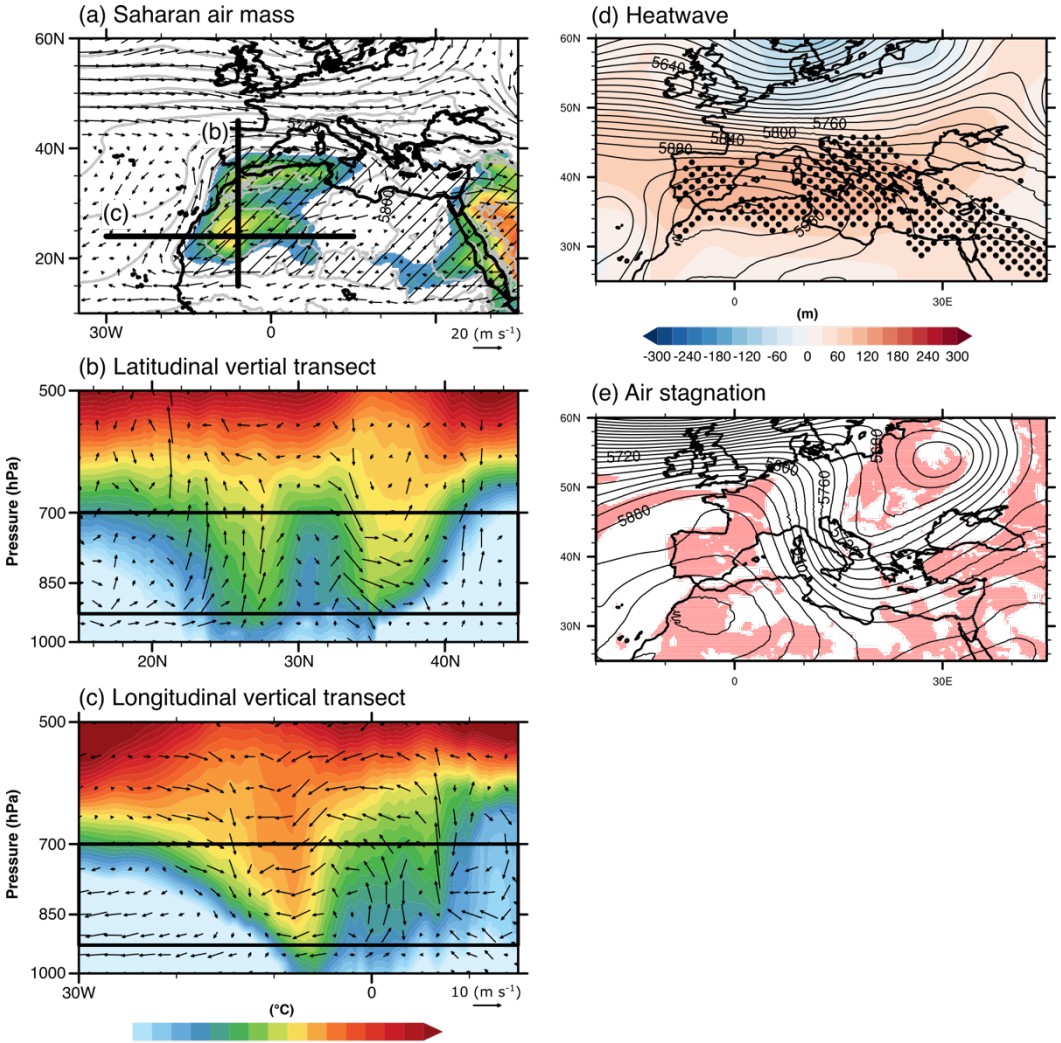

**Figure 9:** Meteorological conditions at the beginning of the ozone outbreak in July 2017. **(a)** Horizontal distribution of mean potential temperature from 925 hPa to 700 hPa and **(b–c)** transects of vertical profiles of potential temperature on 12 July 2017. Black bold straight lines, black arrows, and grey contours in panel (a) indicate the positions of respectively the transects of vertical profiles, daily mean horizontal wind and geopotential height at 500 hPa. Areas featuring 1000–500 hPa layer geopotential thickness is larger than 5800 m are hatched. Black rectangles and arrows in panels (b–c) indicate positions at 925 and 700 hPa, and vectors of horizontal wind and vertical velocity multiplied by 200. **(d)** Areas under heatwave conditions (black dot) on 12 July 2017, with black contour lines and colour shades indicating daily mean and daily anomaly of geopotential height at 500 hPa, respectively. **(e)** Regions under air stagnation conditions (light red shade) and daily mean geopotential height at 850 hPa (black contour lines) on 18 July 2017.

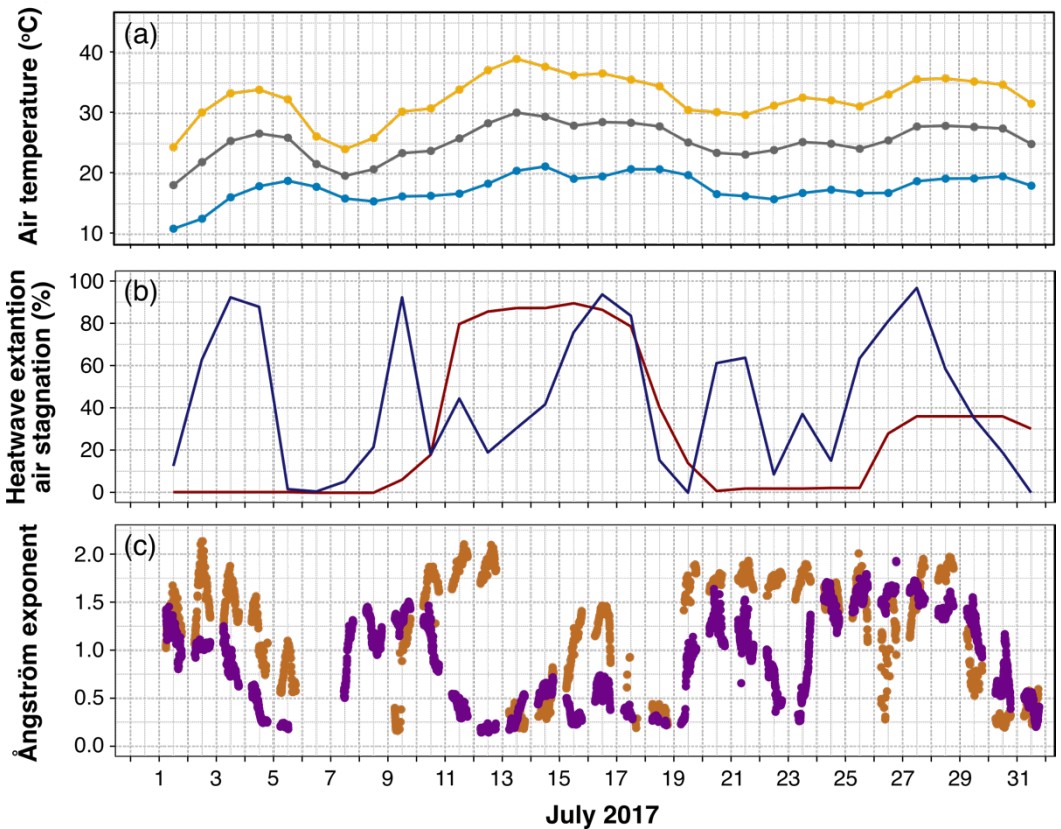

**Figure 10:** Meteorological conditions over the Central Iberian Peninsula (38–42°N, 2–7°W) in July 2017: **(a)** daily maximum (gold), mean (grey), minimum (blue) temperatures, **(b)** fractions of grid which satisfy the criteria for heatwave (dark red) and air stagnation (navy), and **(c)** Ångström exponents in Madrid (brown) and Granada (purple) (c).

The anticyclonic conditions and the arrival of Saharan air over the Iberian Peninsula induce heatwave conditions (see dots in Fig. 9d), that also reach the Eastern Mediterranean. Air temperatures over the Central Iberian Peninsula (38°–42°N, 2°–7°W) start increasing in early July and reach a maximum on 13 July (Fig. 10a). In Madrid, the surface air temperature reaches 40.6 °C on 13 July (Kew et al., 2019). The heatwave prevails over almost all the Central Iberian Peninsula from 11 to 18 July (dark red line in Fig. 10b). This situation usually favours air subsidence, clear skies, and solar heating.

Typically, concomitant with anticyclonic conditions and heatwaves, air stagnation is an additional factor favoring the production and accumulation of air pollutants. This is also the case over the Iberian Peninsula on 15–17 July 2017 according to the air stagnation indicator shown in Figures 9e and 10b (navy line). The anticyclone over Northern Africa likely plays a key role for air stagnation, as it blocks the westerly flow, decreasing both near-surface and upper-air wind speed, and reducing precipitations. This situation continues until 18 July.

### 3.3.2 Photochemical production of ozone from biogenic emissions

These meteorological conditions remaining for over one week favour photochemical production of ozone and likely contributes to trigger the ozone outbreak over the Iberian Peninsula. On 16 July 2017, an enhancement of ozone concentration over the Iberian Peninsula is observed by IASI+GOME2 (red rectangle in Fig. 3a). The ozone plume is co-located with low CO concentrations for the period from 15 to 17 July (indicated in Fig. 3a–c and red line in Fig. 8c from TCR-2 reanalysis), indicating minor influence of combustion-related emissions (typically from biomass burning and some anthropogenic

activities). TCR-2 show an ozone plume over the Canary Islands on 12 July (not shown), which is transported along the edge of warm Saharan air, for arriving to Western Iberian Peninsula on 15–16 July (Fig. 3b). In addition, TCR-2 shows ozone plumes originating from North America that are transported across the North Atlantic for arriving to North-western Iberian Peninsula on 13–15 July (not shown). However, those ozone plumes transported to the Iberian Peninsula are only partially depicted by IASI+GOME2 observations. These differences could be associated with misrepresentation of these ozone plumes

in the simulations or lack of sensitivity (or spatial coverage) in the satellite data (particularly over the ocean). This last aspect is related to a reduction of the sensitivity of IASI+GOME2 over the ocean, due to smaller thermal contrasts than over land (Cuesta et al., 2013). The 7-day back-trajectories confirm that the air masses come from the west over the North Atlantic and offshore transport from Western Africa in the middle troposphere (Fig. 3f). It suggests that the link between those ozone plumes and emissions in North America is unclear. Pay et al. (2019) showed that ground-level ozone concentrations over the

Iberian Peninsula were strongly affected by vertical mixing of ozone-rich layers during a high-ozone event in July 2012. The ozone concentration level over the Iberian Peninsula (over 70 ppb; Fig. 3a) is compatible with that in the middle troposphere (e.g., Petetin et al., 2016).

Tropospheric ozone concentrations near the Iberian Peninsula can be affected by subsidence of ozone-rich air masses from the upper to the lower troposphere, as often found during anticyclonic conditions over the Mediterranean basin during summer

(e.g., Akritidis et al., 2016; Doche et al., 2014; Kalabokas et al., 2013; 2020; Zanis et al., 2014). Figure 11 describe these aspects in terms of meteorological conditions and middle/upper tropospheric ozone on 15 July. Low-pressure systems are observed over Eastern Europe while anticyclonic conditions prevail over Western Europe and the Atlantic (Fig. 11c–d). Some traces of strong subsidence are remarked over France and north of the Iberian Peninsula, between the high- and low-pressure systems, as shown by positive omega vertical velocity at 700 hPa (Fig. 11f, and also at 500 hPa, not shown). Some subsidence

is also seen over the Atlantic, west of the Iberian Peninsula. Similar subsidence conditions have already been observed during regional ozone episodes over the western Mediterranean in spring and summer (Kalabokas et al., 2017; 2020). IASI+GOME2 ozone columns from 6 km to 12 km in Dobson units (DU) are relatively enhanced (~25–30 DU) over Northern and part of Central Iberian Peninsula (Fig. 11b). Relatively larger concentrations of ozone are also simulated by TCR-2 over the Iberian Peninsula at 500 hPa but only the northern part at 700 hPa (respectively ~65 ppb and ~80 ppb, Fig. 11c–d). A transect of

vertical profiles of ozone from TCR-2 along the latitude of 41°N clearly shows downward transport of ozone-rich air masses around 5–10°W (Fig. 11e), but east of 5°E the downward transport of ozone is rather limited below 700 hPa. On the other

hand, IASI+GOME2 retrievals at the lowermost troposphere (Fig. 11a) clearly depict enhanced ozone concentrations over Southern Iberian Peninsula and east of 5°E over land. Therefore, this southern part of the LMT ozone plume is probably not associated with downward transport of upper tropospheric ozone, as it is not colocalised with the region affected by subsidence (i.e. Northern Iberian Peninsula and the Atlantic west and north of the peninsula). Local production of ozone is also suggested by a moderate enhancement of ozone seen at surface level at 3°W of the transect of TCR-2 profiles (Fig. 11e).

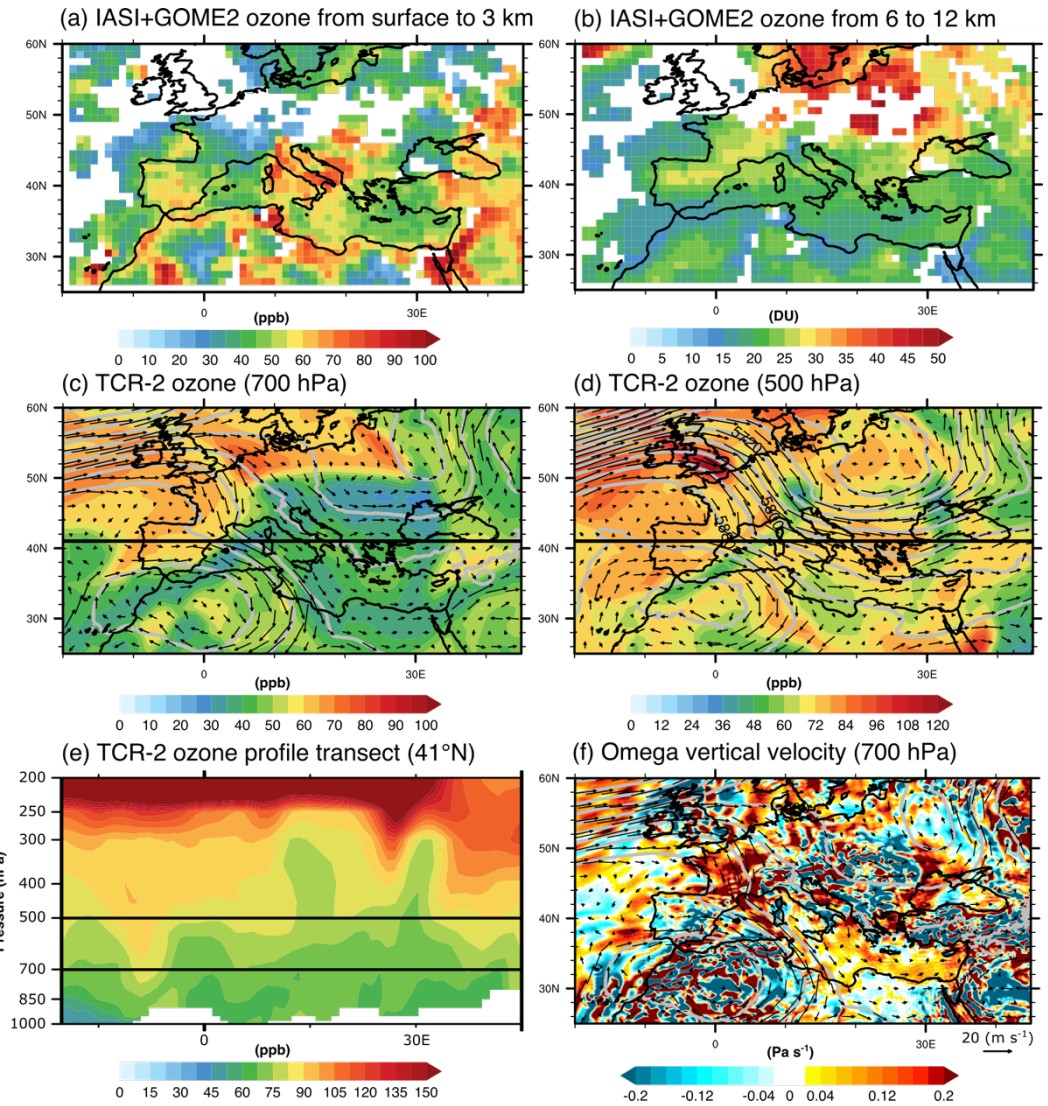

**Figure 11:** Distributions of the meteorological conditions and the middle/upper tropospheric ozone over Europe and the surrounding areas on 15 July 2017. Horizontal distributions of IASI+GOME2 lowermost tropospheric ozone from surface to 3 km **(a)** and upper tropospheric ozone column from 6 km to 12 km **(b)**, TCR-2 ozone at 700 hPa **(c)** and 500 hPa **(d)** and omega vertical velocity at 700 hPa **(f)** from ERA5. **(e)** Transect of vertical profile of TCR-2 ozone at 41°N. Black bold straight

lines in panels (c–d) indicate the positions of the transect of vertical profile. Winds and geopotential height are indicated by black arrows and grey contour lines in panels (c, d and f).

Meanwhile, enhanced concentrations of ozone precursors like $NO_2$ and HCHO are shown over Southern and Central Iberian Peninsula by the tropospheric reanalyses (Fig. 3d–e). These precursors are likely emitted locally, as their short lifetimes prevent the direct influence of long-range transport. The highest levels of $NO_2$ are seen over Madrid (the largest Spanish urban area) and northwest from it (León and Palencia provinces), which is a region with significant mining and power generation industries

(Cuevas et al., 2014). The area with the highest levels of HCHO, which is the most important intermediate compounds in the degradation of VOCs in the troposphere, extends over most of Spain (except for the south-eastern region). In addition, low wind speed conditions may allow the accumulation of these ozone precursors.

Figure 12 shows anthropogenic and biogenic emissions of ozone precursors. Biogenic VOCs (BVOCs), dominated by isoprene and monoterpenes, are especially important for ozone production since they are usually highly reactive. According to the

480 MACC-MEGAN emission inventory in July 2017, biogenic emissions of isoprene and monoterpenes are most abundant over the Mediterranean basin and moderate levels of monoterpenes over Central and Eastern Europe (see Fig. 12c–d). This kind of biogenic emissions can significantly increase by high temperatures (Curci et al., 2009), as those registered during the heatwave in July 2017. The impact of BVOCs on ozone production is likely important over areas with major sources of $NO_X$, and thus low VOC-to-$NO_X$ ratios, as over the Po Valley, North-eastern Spain and Central Europe (Castell et al., 2008; Curci et al.,

2009; Sartelet et al., 2012). Thus, we expect that the ozone plume over the Iberian Peninsula is mainly associated with local biogenic emissions of VOCs collocated with large emissions of $NO_X$. Sources of this last compound are probably soil related. According to CAMS-GLOB-ANT and CAMS-GLOB-SOIL emission inventories (Fig. 12), the amounts of anthropogenic emissions of $NO_X$ and soil emissions of NO are respectively 6.8 Gg and 5.8 Gg over Central Iberian Peninsula (38–42°N, 2–7°W) in July 2017. The relative contribution of soil NO emissions to the total tropospheric $NO_X$ budget would not be negligible,

although a qualitative estimation would be difficult due to the large uncertainties for their quantification (e.g., Hudman et al., 2012; Vinken et al., 2014). Relatively low CO abundances also suggest that combustion emissions are not dominant.

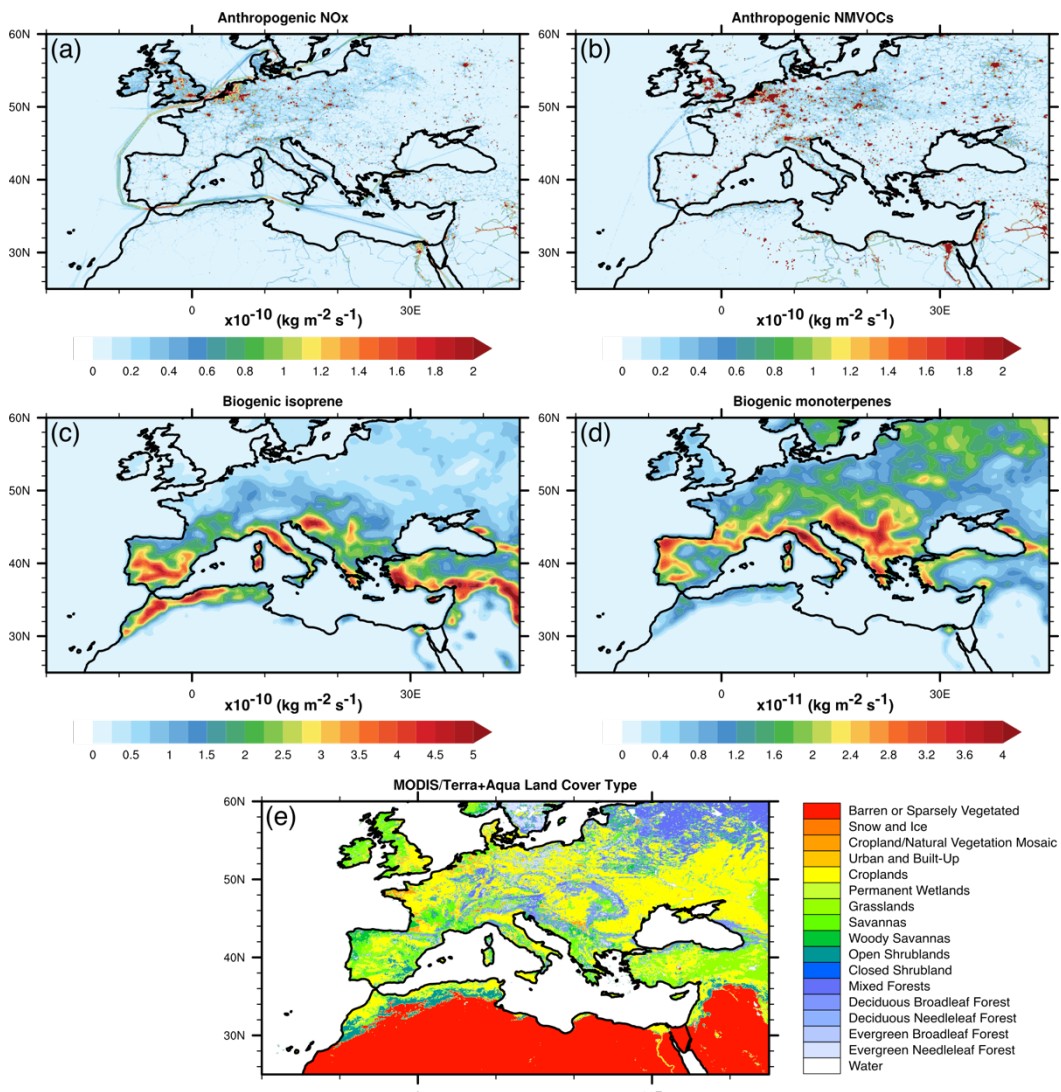

**Figure 12:** Horizontal distributions of anthropogenic and biogenic emissions, and land cover types over Europe and the surrounding regions. Monthly mean emissions of **(a)** anthropogenic $NO_X$ and **(b)** anthropogenic NMVOCs from CAMS-GLOB-ANT, and **(c)** biogenic isoprene and **(d)** biogenic monoterpenes from MEGAN-MACC in July 2017. **(e)** MODIS land cover types in 2017.

The current analysis suggests significant concomitant roles of the heatwave, air stagnation and high emissions of BVOCs contributing to the ozone outbreak over the Iberian Peninsula. However, it is difficult to quantitively evaluate the influence of each of them. Previous studies reported significant correlations between surface ozone abundance and the percentage of air stagnation in the Iberian Peninsula and other regions in Europe (Garrido-Perez et al., 2018; 2019), and in North America (e.g., Leibensperger et al., 2008). On the other hand, other studies over the United States suggest a minor role of high temperatures,

since these conditions lasting several days do not result in a significant ozone increases in any region of this country (Sun et al., 2017).

## 3.4 Impact of anthropogenic emissions in Western and Central Europe

### 3.4.1 Local origin of an ozone plume in Western Europe

The southernmost ozone plume formed over the Iberian Peninsula splits into two branches on 18 July. The northern branch of the plume is transported over Southern France (Fig. 4e and red rectangle in Fig. 5a). An additional dense ozone plume can also be identified over Eastern France by IASI+GOME2 (black rectangle in Fig. 5a), in agreement with surface in situ measurements (not shown). In this region, TCR-2 also simulates moderately high ozone concentrations at 850 hPa (black rectangle in Fig. 5b). The tropospheric reanalysis also depicts a denser ozone plume over Northern Spain, Western France and the nearby Atlantic. This last one is not depicted by IASI+GOME2 at the LMT nor by in situ measurements at the surface. Ozone concentrations simulated at surface level (not shown) are rather moderate at this location, but higher over Eastern France, thus in better agreement with satellite data. At the upper troposphere (between 6 and 12 km of altitude, not shown), IASI+GOME2 does depict an ozone plume over Western France and located slightly north of the one simulated by TCR-2. This suggests uncertainties in the vertical and horizontal location of this lofted ozone plume in the model. A limitation of TCR-2 may come from the fact that the satellite ozone observations assimilated in TCR-2 are derived from TES measurements with a coarse horizontal resolution (only nadir pointing) and most sensitivity to ozone at the free troposphere at lowest.

The middle latitude ozone plume observed over Eastern France on 18 July (black rectangle in Fig. 5a) may partially originate from ozone and/or ozone precursors from the Iberian Peninsula, namely the northern branch of the ozone plume, according to forward trajectories from the previous day (not shown), which are similar to those on 18 July (Fig. 4e). However, this ozone-enriched air masses are co-located with clearly higher CO, $NO_2$ and HCHO concentrations (black lines in Figs. 8c–e) than those observed over the Iberian Peninsula the day before (red lines in Fig. 8c–e). These air pollutants are associated with a dense plume of CO, $NO_2$ and HCHO simulated by TCR-2 over Central Europe (including Eastern France in Fig. 8c–f). This suggests the influence of additional sources of these atmospheric constituents (whose evolution is shown by black lines in Fig. 8c–e). Anthropogenic emissions of $NO_X$ and particularly NMVOCs clearly show hotspots over Central Europe (with populated urban areas and road transportation), according to the CAMS-GLOB-ANT inventory (Fig. 12a–b). Moderately high BVOC emissions (particularly monoterpenes, Fig. 12d) are also seen over Eastern France and larger ones over Southern France. These ozone precursors are transported westwards by easterly winds under anticyclonic conditions. We conclude that the ozone plume over Eastern France on 18 July is highly influenced by anthropogenic emissions from Central Europe, although not particularly from Eastern France itself (exhibiting lower anthropogenic emissions).

In the following days, the concentrations of ozone and its precursors within the middle latitude plume in Central Europe gradually increase (black line in Fig. 8a–b), while being transported eastwards (Fig. 4c–d). When the polluted plume passes over Northern Italy and Germany (not shown), it encounters large local anthropogenic emissions of precursors that probably

mix within the plume. On 21 July, this polluted plume reaches Eastern Europe (extending from Hungary to Eastern Ukraine, see black rectangle in Fig. 6a).

### 3.4.2 Transatlantic transports from North America

Long-range transatlantic transport of air pollutants and their impact on air quality over Europe have been previously examined by using chemistry transport models, satellites, aircrafts and surface observations (e.g., Guerova et al., 2006; Li et al., 2002; Val Martin et al., 2006). Guerova et al. (2006) estimated that the contribution of ozone from North America to Europe ranged from 3–5 ppb in the planetary boundary layer and 10–13 ppb in the middle and upper troposphere in summer.

During the analysed period in July 2017, an ozone plume is transported across the Atlantic from North-western United States and reaches the eastern part of the British Islands on 17 July (according to both IASI+GOME2 and TCR-2 data, which are not shown). This is supported by 7-day back-trajectories from HYSPLIT (Fig. 5g). This northernmost plume shows rather moderate concentrations (~53 ppb, blue line in Fig. 8a), and we expect limited impact on the surface ozone concentrations over Europe. We describe this ozone plume here as part of the moderate ozone outbreak, combining transatlantic and local contributions.

The ozone plume coming from the Atlantic is observed on 18 July over Northern Germany by IASI+GOME2 (blue rectangle in Fig. 5a). This is a known hotspot of anthropogenic emissions (Fig. 12a–b). At this stage, these air masses exhibit low CO concentrations but are relatively rich in $NO_2$. During transport over Northern Europe from 17 to 20 July, the concentrations of ozone, CO, $NO_2$ and HCHO within this plume gradually increase (blue lines in Fig. 8). Production of ozone during transport is likely enhanced by mixing with local anthropogenic emissions of its precursors.

Meanwhile, a large and dense CO plume (but poor in ozone) is depicted by the IASI satellite (Fig. 5d) over the North Sea, which is located north of the ozone plume shown by IASI+GOME2 over Northern Germany (Fig. 5a). This CO plume originates from Central Canada, according to 7-day back-trajectories (Fig. 5h). In this location, TCR-2 reanalyses depict rather weak increases in background concentrations of CO at 850 hPa between the east coast of England and Netherland (Fig. 5c). The simulated plume extends vertically from the lower troposphere (~930 hPa) to the upper troposphere (~240 hPa) with relatively uniform concentration (~110 ppb), thus the total column of CO is larger than the surroundings (as seen by IASI) but not so clearly at a given atmospheric level. Meteorological analyses from ERA5 suggest that the long-range transatlantic transport of these ozone and CO plumes over Northern Europe (~55°N) is likely linked the circulation patterns associated with the Azores anticyclone during the event (see Fig. S2). Similarly, Guerova et al. (2006) found that the frequency of these events depends on the position and strength of the Azores anticyclone.

Some traces of strong subsidence are observed over Central Europe as shown by positive omega vertical velocity at 700 (Fig. S3f) and 500 hPa (not shown) on 20 July. TCR-2 ozone at 700 hPa and 500 hPa show a high concentration ozone belt over Norway, Denmark, Poland and Ukraine (Fig. S3c–d), whereas IASI+GOME2 upper tropospheric ozone columns (from 6 km

to 12 km) does not show high values at those locations but north of them (Fig. S3b). Vertical profile of TCR-2 ozone at 51°N shows a downward transport of ozone-rich air masses around 15–20°E (Fig. S3e). That is the place where the northernmost plume is located (Fig. 4b). Therefore, the northernmost plume might be affected by downward transport from the middle troposphere.

## 3.5 Wildfire and biogenic emissions in the Balkan Peninsula

The southern branch of the southernmost ozone plume formed over the Iberian Peninsula at the beginning of the event is transported eastwards over the Western Mediterranean (on 18 July, Fig. 4e and red rectangle in Fig. 5a), exhibiting low CO, $NO_2$ and HCHO concentrations (red lines in Fig. 8c–e). During the following days, the abundances of these pollutants enhance (Fig. 8c–e) while being transported over Italy and arriving to the Balkan Peninsula on 21 July (Fig. 4f and 6a). Meteorological conditions over Eastern Europe, Italy, and Northern Africa are characterized by extended high pressure, which is accompanied by air subsidence, clear skies, and intense solar heating (Fig. 6f). Heatwave conditions are found over Southern Italy and the Western Balkan Peninsula. This is favourable for ignition and spread of wildfires as well as enhancement of biogenic emissions, as typically occurs over the Mediterranean basin during summer (e.g., Turco et al., 2017). According to MODIS Fire Radiative Power data, wildfires are clearly visible over Central and Southern Italy, and the coast of the Adriatic Sea in the Balkan Peninsula (Fig. 6g). Significant enhancements of CO are seen at this last location (Fig. 6c), probably originating from local wildfires of savannas (the dominant land cover there, Fig. 12e). Moreover, dense plumes of $NO_2$ and HCHO are simulated over the Balkan Peninsula, from the Adriatic coast and inlands further east (Fig. 6d–e). These regions are characterized by large biogenic emissions (Fig. 12c–d). Meanwhile, low dispersion conditions under the stagnating anticyclonic conditions allow the accumulation of $NO_2$. Therefore, two sources of ozone precursors are found in this region: wildfires along the Adriatic coast and inland biogenic emissions further east.

Three-days forward-trajectories suggest that a part of the air masses over Eastern Italy move southwards following northerly winds (Fig. 4f–g). Air pollutants likely formed over the Balkan Peninsula are then transported to the Eastern Mediterranean and Northern Africa (Fig. 6a–e). These results suggests that air quality over the Mediterranean may be influenced by eastward transport of ozone pollution from Western Europe and subsequent photochemical production. It is worth noting that previous modelling studies highlighted that regional sources of $NO_X$ and BVOCs are the most important factors controlling summertime ozone in the lower troposphere over the Mediterranean (Richards et al. 2013). The role of transport of air pollutants over the Mediterranean has also been previously remarked. Indeed, the summer circulation over the Eastern Mediterranean is dominated by persistent northerlies known as the Etesians (e.g., Tyrlis et al., 2013). The Etesians are known to have a significant impact on air quality over the Eastern Mediterranean as they transport anthropogenic emissions from European industrial areas (Kalabokas et al., 2008; Lelieveld et al., 2002) and the biomass burning emissions from countries bordering the Black Sea and Southern Europe (de Meji and Lelieveld, 2011; Hodnebrog et al., 2012; Sciare et al., 2008). The Etesians cause the vertical downward transport of ozone-rich air masses from the upper troposphere and the lower stratosphere as well as the horizontal

transport of polluted air masses (Doche et al., 2014; Kalabokas et al., 2007; 2013; Zanis te al., 2014). The strength of Etesian winds and of the subsidence over the Mediterranean basin are at a maximum in July (Tyrlis and Lelieveld, 2013).

## 3.6 Agricultural burning emission in the north coast of the Black Sea

The three ozone plumes previously monitored while travelling eastwards across Europe (Figs. 4, 6–7) mix into a single large ozone plume that reaches the northern coast of the Black Sea (Romania, Moldavia, Ukraine, and Russia) on 23 July (Fig. 7a–b). Over this region, we can see strong enhancements of CO and relatively weak concentrations of $NO_2$ and HCHO (Fig. 7c–f). Active biomass burning spots are clearly visible over this region (Fig. 7f), which is mainly covered by croplands (as shown by Figure 12e). These fires probably correspond to agricultural burning, which is a common practice over Eastern Europe and European Russia for removing crop residuals for new planting or clearing weeds and brushes for grazing. This is the single largest cause (an average of 86%) of the fires detected in Ukraine in 2001–2003 (Korontzi et al., 2006). These events typically occur in July and August, after the harvest of wheat, rye and barley (planted during the previous winter or spring for the last one).

The presence of significant local sources of CO, much likely linked to agricultural burning, are suggested by an increase of its concentration within the polluted plume from ~118 ppb to ~134 ppb during the day of 23 July (Fig. 8c).  These local emissions likely mix with the pollutants transported from Western Europe. On other hand, simulated abundances of HCHO and $NO_2$ do not exhibit remarkable changes during this day (Fig. 8d–e). During the following days, the pollution plume is transported away eastwards from the fire hotspot while the concentrations of ozone precursors gradually decrease. The temporal variation of ozone observed by IASI+GOME2 within the plume show different features with respect TCR-2 reanalyses (see Fig. 8a–b). The satellite approach depicts gradual enhancements of ozone during transport away from the Black Sea coast. In absence of other sources or ozone advection, this increase in ozone concentrations is likely linked to photochemical production. On the other hand, TCR-2 simulates a temporal decrease of ozone concentrations for this plume. The differences between the model and the satellite data are probably associated with the representation of biomass burning emissions and its impact on the simulation of ozone production. Simulations of these processes and their quantification are currently challenging, as shown by the large discrepancies between different state-of-the-art global tropospheric ozone reanalysis products (e.g., Huijnen et al., 2020). Near-surface ozone concentrations are weakly and mainly indirectly constrained by the satellite observations used for assimilation in reanalysis such as TCR-2. In addition, only very few ground-based stations monitor air pollutants over Eastern Europe, as their geographical coverage is far from homogenous across the globe. Particularly in those locations, satellite observations of lowermost tropospheric ozone as those derived by IASI+GOME2 are particularly appreciated for filling the gap of ground-based observations.

## 4 Conclusions

We have presented a detailed study of the daily evolution and associated sources of precursors of a moderate ozone outbreak transported across Europe in July 2017, by using IASI+GOME2 multispectral satellite observations and tropospheric chemical reanalysis TCR-2. The multispectral satellite approach offers the currently unique capacity of observing the ozone distribution in the lowermost troposphere (below 3 km a.s.l.), which is in fair consistency with surface observations. This regional to continental analysis cannot be made with in situ observations only since their spatial coverage is limited. This stresses the contribution of satellite-based approaches for observing ozone pollution provided their highly valuable horizontal coverage.

The moderate European ozone outbreak analysed here is associated with several sources of ozone precursors: biogenic, anthropogenic and biomass burning emissions. We describe the sources of precursors and the transport pathways of these ozone plumes (see scheme in Fig. 13), using IASI+GOME2, TCR-2 and other datasets. At the beginning of the event, warm Saharan air masses are advected northwards over the Iberian Peninsula by the circulation associated with an anticyclone over Morocco, and then heatwave conditions prevailed over the Iberian Peninsula. There, an ozone plume (southernmost plume) is formed on 16 July 2017, as temperature-induced biogenic emissions increase and collocated high anthropogenic emissions. This ozone plume is co-located with low CO concentrations, but high $NO_2$ and HCHO abundances. Then, the southernmost plume is transported eastward across the North-western Mediterranean, exhibiting low concentrations of CO, $NO_2$ and HCHO. These last ones increase as the plume approaches Italy and remain high while reaching the Balkan Peninsula. Over this region, two kinds of sources of ozone precursors are found: wildfire emissions along the coast of the Adriatic Sea and biogenic emissions in the inland region of the Balkan Peninsula.

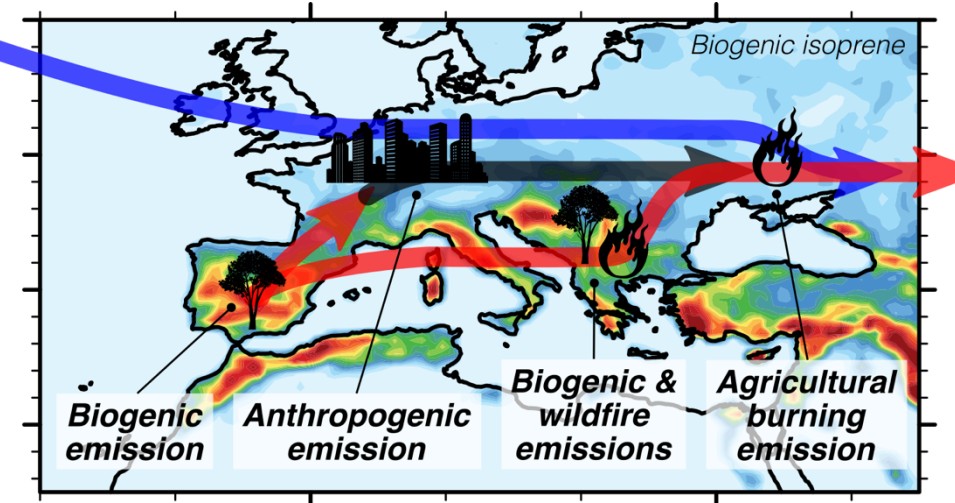

**Figure 13:** Schematic of mechanisms that control the moderate ozone outbreak in July 2017 across Europe.

Meanwhile, two other ozone plumes (middle latitude and northernmost plumes) are also observed over Western and Central Europe. The northernmost plume was originally formed over Northern United States and transported across the Northern Atlantic, whereas the middle latitude plume over Eastern France. Both pollution plumes show high concentrations of ozone precursors, especially $NO_2$, suggesting a significant impact of anthropogenic emissions from Central Europe. These ozone plumes and the one located over the Balkan Peninsula are transported eastwards and mix gradually into a single large plume which reaches the north coast of the Black Sea. There, active agricultural burning provides with large abundances of precursors, as clearly remarked for CO. The satellite approach highlights significant photochemical production of ozone in the following days, as the polluted air masses are transported further east. Chemical reanalyses did not reproduce this last aspect, which occurred in a remote region and subject to large uncertainties in biomass fire emissions.

European anthropogenic emissions of primary air pollutants have been controlled and dropped considerably since the 1990s. Therefore, the contribution of biogenic and biomass burning emissions as precursors of tropospheric ozone is expected to play a significant role locally and on intercontinental transport. Along with global warming, unusually long and intense heatwaves have been become more frequent and severe over Europe (Russo et al., 2015). This is clearly seen in climate model projections at regional and global scales (e.g., Seneviratne et al., 2021). In addition, summer drought conditions and high temperatures are primary drivers of the inter-annual variability of fires (e.g., Turco et al., 2017). It is worth noting that many model intercomparisons have suggested that anthropogenic emissions are harmonised, but biogenic emissions usually differ from one model to the other (e.g., Im et al., 2015). Generally, state-of-the-art biogenic emissions inventories such as MEGAN tend to overestimate isoprene emissions especially in Scandinavian countries and southwest Europe and underestimate those of monoterpene (e.g., Jiang et al., 2019). It is important for policy and decision making to better observe and understand the response of ozone to change of emissions. Satellite-based approaches can contribute to the quantification of these photochemical processes and highlight remaining gaps in modelling tools, thus their synergism may allow a better understanding air quality degradation and improving the efficiency of pollution mitigation policies.

*Acknowledgements*

Authors acknowledge the financial support of Centre National des Etudes Spatiales (CNES, the French Space Agency) via the SURVEYPOLLUTION project from TOSCA (Terre Ocean Surface Continentale Atmosphère), the Université Paris Est Créteil (UPEC), and the Centre National des Recherches Scientifiques–Institut National des Sciences de l'Univers (CNRS-INSU), for helping in achieving this research work and its publication.

We thank the French atmospheric data centre AERIS (https://www.aeris-data.fr, last access: 10 June 2022) for supporting the production of IASI+GOME2, data access and computing resources provided by the ESPRI IPSL mesocentre (https://mesocentre.ipsl.fr/, last access: 1 April 2022) and the European Air Quality e-Reporting for in situ surface measurements of ozone, EUMETSAT for GOME-2 level 1 data (provided by the NOAA CLASS data portal). IASI is a joint mission of EUMETSAT and CNES. We thank the Institut für Meteorologie und Klimaforschung (Germany) and RT Solutions

(USA) for licenses to use, respectively, the KOPRA and VLIDORT radiative transfer models. We also thank Zhaonan Cai from the Chinese Academy of Sciences (China) and Xiong Liu from the Harvard-Smithsonian Center for Astrophysics (USA) for their help to produce IASI+GOME2 data. The Programme National de Télédétection Spatiale (PNTS) and the Agence Nationale de la Recherche (ANR) also contributed in supporting the development of the IASI+GOME2 satellite product. Part of this work was conducted at the Jet Propulsion Laboratory, California Institute of Technology, under contract with the National Aeronautics and Space Administration (NASA).

We warmly acknowledge all datasets provided for this study: CO satellite retrievals from IASI by ULB/LATMOS (Université Libre de Bruxelles/Laboratoire Atmosphères, Milieux, Observations Spatiales) laboratories (special thanks to C. Clerbaux and J. Hadji-Lazaro), the Terra and Aqua MODIS active fire products by NASA's Fire Information for Resource Management System (FIRMS) (https://earthdata.nasa.gov/firms), part of the NASA Earth Observing System Data and Information System (EOSDIS), MACC-MEGAN, CAMS-GLOB-ANT and CAMS-GLOB-SOIL emission inventories by the Emissions of atmospheric Compounds and Compilation of Ancillary Data (ECCAD) (https://eccad.aeris-data.fr/), the global land cover data by the Land Processes Distributed Active Archive Center (LP DAAC) (https://lpdaac.usgs.gov/), and the aerosol optical depth level 2 data by the Aerosol Robotic Network (AERONET) project (https://aeronet.gsfc.nasa.gov/). We thank Lucas Alados Arboledas, Jose Maria San Atanasio, and Ana Diaz Rodriguez for their effort in establishing and maintaining Granada and Madrid sites of AERONET. We gratefully acknowledge the NOAA Air Resources Laboratory (ARL) for the provision of the HYSPLIT transport model used in this publication.

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
