# Peer review of "Impact of different sources of precursors on an ozone pollution outbreak over Europe analysed with IASI+GOME2 multispectral satellite observations and model simulations"

_Atmospheric Chemistry and Physics, 2022_

## Author Comment (AC1)

**Response to Reviewers of Manuscript**

"Impact of different sources of precursors on a high-ozone event over Europe analysed with IASI+GOME2 multispectral satellite observations and model simulations" by S. Okamoto et al.

The authors are grateful to the reviewers for the time and thought that they clearly put into their reviews and comments regarding our paper. We have addressed all their comments below and accordingly changes into our revised manuscript, which has led to substantial improvement in clarity and completeness. The original comments from the referees are in black and our responses in blue. Please see below our point-by-point replies to the comments of the reviewers.

**Reviewer 1**

**Comments:**

Okamoto et al. present in their paper an analysis of ozone plumes in the lowermost troposphere and follow those plumes through Europe using a new satellite product IASI+GOME2. They use a chemical reanalysis TCR-2 to investigate the chemical composition and attribute sources to the plumes.

The topic of the paper is important and certainly within the scope of ACP. However, the authors need to revise the structure and logic of their manuscript (see general points). Also, it is important to motivate the usage of TCR-2 data, which often is not in agreement with validated measurements, such as IASI+GOME2 ozone, or IASI CO. In this form, the conclusions drawn from TCR-2 are not very sound. I encourage the authors to address my points below in a major revision of the manuscript.

Thank you very much for your thorough and helpful review of our paper. Please see below for specific responses to your concerns.

**General Comments:**

- Overall, the structure could be improved to help the reader to understand the contents: When introducing a new figure, only parts of it are sometimes mentioned in the text and others not, while references to later figures are mixed in the same part of the manuscript. I would suggest to first think, which point should be made with a certain figure, then compile the figure with all the information necessary to make the point, and introduce the whole figure at once, in order to make the point then in the text. Later in the discussion, information from different figures can be put together, but in the beginning, it would be preferable to introduce the figures in the same order as they appear in the manuscript.

Reply 1: Done. We have added sentences to introduce each figure as a whole, before using it for the analysis and discussion. Additionally, we have changed the order of the figures according to your comment after reply 3.

- It is not clear to me, why the validation approach of the satellite data with ground

based in situ measurements is performed on such a coarse spatio-temporal grid.

Reply 2: Clarified. In the present paper, we compare IASI+GOME2 lowermost tropospheric ozone retrievals averaged in space at 1° × 1° resolution with respect to daily time-averaged in situ measurements at surface level. Very similar coincidence criteria and spatio-temporal grid has been used in the comparisons of IASI+GOME2 satellite retrievals and surface in situ data done over East Asia by Cuesta et al., (2018) and over Europe by Cuesta et al., (2022). These three comparisons depict how the satellite data captures surface ozone variability and how they represent near-surface ozone abundances. Since these two kinds of observations (satellite and in situ) are not sensitive at the same height, these comparisons are not meant to be "apples-to-apples" validations. Indeed, IASI+GOME2 retrievals of lowermost tropospheric columns of ozone (below 3 km of altitude) show a relative maximum of sensitivity at about 2.2 km of altitude over land in mid-latitudes (see Cuesta et al., 2013). At this altitude and during the overpass time of the MetOp satellites around 09:30 local time, the IASI+GOME2 approach likely measures ozone concentrations at the residual atmospheric boundary layer. We expect that the variability of these last ones is better represented by daily or afternoon averages than morning surface concentrations. Indeed, these last ones have not yet been mixed vertically within the whole boundary layer.

Additionally, spatial averaging of satellite data reduces its random errors. We have chosen a spatial resolution of  $1^{\circ} \times 1^{\circ}$  for IASI+GOME2 data as a good compromise for limiting its errors while depicting the main ozone spatial variability at daily scale of regional scale ozone pollution outbreaks.

These aspects are clarified in the revised manuscript (RM) as (line 102-108) "For reducing random errors, the dataset is averaged in a regular horizontal grid of 1° × 1°, in the same way as done by Cuesta et al. (2018). This spatial resolution enables the observation of the horizontal distribution of regional-scale ozone plumes, such as those analysed in the current study over Europe. Clear sky conditions ensure good daily coverage of satellite data, as was the case during the analysed event. Ozone concentrations from the surface to 3 km of altitude (a.s.l.) are provided as an average ozone volume mixing ratio in ppb within the layer, which is calculated as the ratio of lowermost tropospheric partial columns (in molecules per square centimetre) of ozone and air. Hereinafter, this amount is designated as IASI+GOME2 LMT ozone concentration.", and (line 124–132) "We consider afternoon averages (12:00–16:00 local time - LT) of these surface concentrations that are expected to be vertically mixed within the mixing atmospheric boundary layer. This is probably more comparable to the IASI+GOME2 retrievals than morning surface concentrations that have not been mixed within the whole boundary layer (Cuesta et al., 2022). This is explained by the fact that the IASI+GOME2 LMT retrievals are sensitive around ~2.2 km of altitude during summer over land (see Cuesta et al., 2013), thus mostly measuring ozone concentrations at the residual atmospheric boundary layer. It is worth noting that as for other ozone satellite retrievals, the height of maximum sensitivity of IASI+GOME2 vary depending on the observing conditions. Due to the use of thermal infrared measurements, the height of sensitivity over the ocean is 1 to 2 km higher than that over land (as the thermal contrast

between the near-surface air and the surface itself is weaker over the ocean than over land).".

- Section 3 should be reorganized: First the observed plumes should be introduced, then trajectories connected to these plumes should be discussed. Figure 7 shows the IASI+GOME2 measurements the first time in this paper, but these measurements are motivating the study after all. So, I would highly recommend to start with this figure and explain it first. Then either explain the meteorological situation, trajectories or chemical reanalysis results. Or maybe even start with a figure containing several panels of IASI+GOME2 measurements for different days? This would illustrate nicely the temporal evolution of the plume(s), and demonstrate the strength of the observations, which are the base of this work.

Reply 3: Agreed and done. As recommended, we have reordered the figures and the corresponding descriptions to show the ozone plume depicted by IASI+GOME2 at the beginning of section 3. We start the section by introducing the IASI+GOME2 ozone map at the beginning of the pollution event (16 July 2017) as Figure 3. And then three-day forward trajectories (Fig. 4) for explaining its evolution. The following figures are ozone and other species maps on 18 July (Fig. 5), 21 July (Fig. 6) and 23 July (Fig. 7). We agree that now it is easier to follow the temporal evolution of the ozone plumes.

- It is not well motivated, why the TCR-2 model is useful to explain the chemical situation. For temporal and horizontal comparisons in ozone between IASI+GOME2 and TCR-2, there are large differences (see Figures 4a/b, 7a/b, 9a/b, 10a/b, 11a/b), which are not explained sufficiently enough here. Sometimes, plumes are marked in the satellite data by boxes, and these plumes are not reproduced by TCR-2. Further, the only comparison of CO between TCR-2 and IASI shows problems in TCR-2. So how can the authors draw conclusions for origin of the pollution, which are only based on this model data, which seems not to match the observations?

Reply 4: Clarified. In the current work, TCR-2 analyses are used as complementary and very relevant information about the abundance of ozone precursors, as well as ozone concentrations at a specific pressure level (850 hPa). Their combined use with satellite data is not expected to provide a validation of either of the datasets nor straight-forward comparisons. This is because model data is provided at a vertical level and satellite measurements are integrated over a partial (ozone) or total (carbon monoxide) atmospheric columns and they are subject to chances in the height of maximum sensitivity according to the observing conditions. Indeed, IASI+GOME2 ozone data are integrated between the surface and up to 3 km of altitude (that we call the lowermost troposphere) with maximum height of sensitivity which is 1 to 2 km higher over the sea than over land, and also vary significantly for each of these surface types (see Cuesta et al., 2013). Retrievals of carbon monoxide (CO) from IASI used here are integrated over the total column and its maximum sensitivity is at the free troposphere, being higher over ocean as well. These differences between the height at which either ozone or CO concentrations are either observed or modeled partly explain how these two datasets

differ. Uncertainties in these two datasets also partially explain their differences. The labels of the maps in the figures of the previous manuscript were not clear enough. Therefore, we clearly indicate in the labels of each panel of the figures of the RM at which altitude the maps refer.

The most important information provided by TCR-2 data is the abundance of ozone precursors, which is used in semi-quantitatively for identifying its origin. This information together with the satellite observation of ozone are one of the important contributions of the paper. The abundance of ozone precursors near the surface and with full spatial coverage is nowadays mainly provided by chemistry-transport simulations, such as TCR-2. The combined use of satellite and model data for analyzing ozone pollution origins is rather original, as most studies on this topic are only based on model data. Analyzing simulated ozone distributions with fine vertical resolution (at a given pressure level) are complementary with respect to ozone satellite data for pointing out potential uncertainties in the model (e.g., as remarked when ozone precursors are emitted by biomass burning north of the Black Sea).

This is clarified in the RM as (line 181-184, 196-208) "Chemical reanalyses provide comprehensive information on the evolution of atmospheric composition, offering the spatio-temporal full coverage of chemistry-transport models and enhanced precision by assimilation of various satellite observations. Data assimilation provides an estimate of the most likely state of system (Lahoz and Schneider, 2014). As support of the present analysis of the link between ozone precursors and an ozone outbreak over Europe, we use the Tropospheric Chemical Reanalysis version 2 (TCR-2). (...) TCR-2 has a T106 horizontal resolution  $(1.1^{\circ} \times 1.1^{\circ})$  with 32 vertical levels from the surface to 4.4 hPa. Nine of these vertical levels are typically found within the lowermost troposphere (below 3 km of altitude). In this study, we mainly use model data at 850 hPa for describing the abundance of ozone precursors and the pathways of ozone plumes at this atmospheric level. Simulations from 1000 hPa to 200 hPa also allow the investigation of the influence of downward transport from the upper troposphere and the lower stratosphere. The model-derived ozone and CO data is used here without smoothing by averaging kernels of the satellite retrievals (as also done by e.g., Foret et al., 2014; Cuesta et al., 2018). This model data provides information with fine vertical resolution on the spatial distribution at these atmospheric pollutants at specific vertical levels. Therefore, they are complementary with respect to satellite retrievals, that represent the abundances of these pollutants integrated vertically within an atmospheric layer (i.e., the LMT for ozone and the total column of CO, described respectively in subsections 2.1 and 2.2) and their sensitivity may vertically vary depending on the observing conditions. The combined use of model and satellite data allow accounting for the possible uncertainties or limitations of each of them, while providing a more complete description of the spatio-temporal evolution of ozone precursors.".

- For some figures, fonts (on color bars or axes) are very small.

Reply 5: Done. We have enlarged small fonts in the figures (Figs. 3, 5, 6, 7, 9, 12).

**Specific comments:**

- Line 13: Define "IASI" and "GOME2"

Reply 6: Done. They are defined in lines 14–15 of the RM.

- Line 38: "The main sources of tropospheric ozone ...": This is a very long sentence, and it is not quite clear, if "in situ photochemical production through oxidation of CO and CH4" is a main source, or if these two parts should be considered separately. Please try to make this sentence shorter and easier to read.

Reply 7: Done. In the RM, we have rephrased the statement as (lines 38–41): "The main source of tropospheric ozone is in situ photochemical production through oxidation of non-methane volatile organic compounds (NMVOCs), carbon monoxide (CO) and methane (CH4), in the presence of nitrogen oxides (NOx) (e.g., Atkinson, 2000). Stratosphere-to-troposphere transport of ozone also significantly enhances its abundance in the troposphere (e.g., Stohl et al., 2003)."

- Line 101: "Ozone concentrations from the surface to 3 km of altitude (a.s.l.) are provided as volume mixing ratio in ppb (parts per billion)": So, this is rather an average value for the ozone concentration below 3 km altitude?

Reply 8: Agreed and clarified. The volume mixing ratios between the surface and 3 km of altitude are calculated as the ratio of the partial columns of ozone and air in this altitude range. These partial columns are obtained as the sum of molecules of either ozone or air below 3 km of altitude. This quantity is indeed an average mixing ratio within the partial column.

This is clarified in the RM as (lines 106–108) ".. are provided as an average ozone volume mixing ratio in ppb within the layer, which is calculated as the ratio of lowermost tropospheric partial columns (in molecules per square centimetre) of ozone and air.".

- Line 104 and following: What kind of measurements are these surface observations of ozone? Are these in-situ measurements? Which kind of measurement technique has been used? Some background information would be useful here.

Reply 9: Clarified. They are in situ measurements performed by the UV absorption technique.

This is clarified in the RM as (lines 116–117) "In the present study, we use hourly surface data from stations in rural areas, derived from in situ measurements performed using the UV absorption technique."

- Line 119: Could you please motivate the time period for the comparison?

Reply 10: Clarified. The comparison is performed during the European ozone pollution

outbreak analyzed in the current paper (e.g. 15-27 July 2017, including one day before and one after). This period of analysis corresponds to a moderate summer ozone pollution event over Europe, which is very common during this season and over this region. Prevailing clear sky conditions also allow a good daily coverage of satellite observations.

This is indicated in the RM as (line 134) ".. period 15–27 July 2017 (one day before and after the major ozone outbreak analysed in the paper)" and (lines 85–86) "This is a moderate pollution event, which is very common during the summer in this region (e.g., Cuesta et al., 2013; Foret et al., 2014; Kalabokas et al., 2020)" And (lines 105) "Clear sky conditions ensure good daily coverage of satellite data, as was the case during the analysed event."

- Line 118 and following: I did not really understand, which quantities are compared here: Do the authors compare averages over the whole time period? Or are data compared in a finer temporal resolution? Which horizontal resolution of the satellite data is compared to ground measurements? Please indicate a bit more in detail the settings of these comparisons.

Reply 11: Clarified. These comparisons are made at daily scale for spatially co-localized datasets of in situ and satellite measurements. For each day, in situ measurements at surface level are averaged during the afternoon (between 12:00–16:00 LT) to be representative of vertically mixed concentrations within the atmospheric boundary layer which are more comparable with satellite measurements sensitive at lowest around 2 km of altitude (see more details on this aspect in reply 2). They are compared with spatially coincident satellite IASI+GOME2 observations, obtained by merging MetOp-A and MetOp-B satellites datasets and averaging at a horizontal grid of 1° × 1° (which is compatible with the representation of regional-scale ozone plumes and allowing a reduction of random errors of the retrievals at pixel resolution). The correlation coefficient in Fig. 1 were calculated at each station for daily data during the period 15–27 July 2017.

We have added some details of these comparisons (lines 123–125) "Here we only use data from "rural, low elevation" and "rural, high elevation" stations. We consider afternoon averages (12:00–16:00 local time - LT) of these surface concentrations that are expected to be vertically mixed within the mixing atmospheric boundary layer." and (lines 148–152) "Figure 2 shows the scatter plots of daily in situ surface observations from the ensemble of stations within three sub-regions (shown as black rectangles in Fig. 1) with IASI+GOME2 data colocalized in time (at daily scale) and in space (satellite data at 1° × 1° sampled at the location of each station) for the period 15–27 July 2017. These sub-regions are (i) the Iberian Peninsula (39°–42°N, 0°–8°W), (ii) Southern and Central Europe (41°–48°N, 3°W–13°E), and (iii) Central and Eastern Europe (45°–50°N, 10°–25°E). The statistics for each of the three sub-regions are shown in Table 1.".

- Line 123: Please define "p".

Reply 12: We have replaced "p" by "P-value from the test of significance" (line 138) and "P-value" (line 153, and Table 1).

- Line 127: Are these numbers given in ppb considered to be an average bias for the given regions? How are these biases calculated from the correlations shown in Fig. 2?

Reply 13: Clarified. The values in Fig. 2 and Table 1 are calculated for the whole dataset (daily co-localized in situ and satellite data) within each region depicted in black rectangles in Fig. 1: Iberian Peninsula, Southern and Central Europe, and Central and Eastern Europe.

This is better indicated in the RM as (lines 148–152) "Figure 2 shows the scatter plots of daily in situ surface observations from the ensemble of stations within three sub-regions (shown as black rectangles in Fig. 1) with IASI+GOME2 data colocalized in time (at daily scale) and in space (satellite data at 1° x 1° sampled at the location of each station) for the period 15–27 July 2017. These sub-regions are (i) the Iberian Peninsula  $(39^\circ-42^\circ N, 0^\circ-8^\circ W)$ , (ii) Southern and Central Europe  $(41^\circ-48^\circ N, 3^\circ W-13^\circ E)$ , and (iii) Central and Eastern Europe  $(45^\circ-50^\circ N, 10^\circ-25^\circ E)$ ."

- Line 130: "but they may reflect differences in the spatio-temporal representativity of surface in situ measurements and satellite data at daily scale and this particular event.": So maybe, it would make more sense to make this correlation based on a finer temporal-spatial resolution? Why did the authors choose a different approach?

Reply 14: Clarified. The correlations are performed at the scales suited for describing the daily evolution of the ozone plumes travelling across Europe. We have compared the dataset at daily scale, which is the smallest possible temporal resolution, corresponding to that of the satellite data. A 1 x 1° spatial resolution is used for the satellite data, for reducing random errors and representing regional scale ozone plumes. We consider in situ data both at the individual station scale (Fig. 1) and as an average over three European regions (Fig. 2 and Table 1). Please see additional comments in reply 2.

- Line 162: Please define the acronym MIROC-CHASER.

Reply 15: Done. It is defined as "Model for Interdisciplinary Research on Climate-Chemical atmospheric general circulation model for study of atmospheric environment and radiative forcing" (lines 191–192).

- Line 167; "32 vertical levels from the surface to 4.4 hPa.": How many of these layers are in the region of interest of this study (lowermost troposphere below 3 km)?

Reply 16: Clarified. Typically, nine vertical levels are found within the lowermost troposphere. We mainly use model data at 850 hPa for tracking plumes of ozone and its precursors within this atmospheric layer. Simulations from 1000 hPa to 200 hPa allow the investigation of the influence of downward transport from the lower stratosphere. These previous lines are added in the RM (lines 197–201).

- Line 182: Why are you not smoothing by averaging kernel? Are the vertical resolutions of IASI+GOME2 and TCR-2 comparable?

Reply 17: Clarified and Agreed. The use of model data is meant to provide complementary information on the distribution of ozone and its precursors at a given vertical level (i.e., 850 hPa). Smoothing model vertical profiles by averaging kernels would integrate vertically ozone concentrations over roughly 4-5 km (the vertical resolution of the satellite data) and the maximum sensitivity would change depending on the observing conditions (particularly the thermal contrast between the near surface atmosphere and the surface itself). Moreover, as suggested by the reviewer, the convolution by averaging kernels may sometimes not be straight-forward due to differences in the altitude ranges of the two datasets (while satellite retrievals derive ozone profiles from the surface and up to 60 km of altitude, ozone concentrations at the higher part of this altitude range is not simulated in the model). For these reasons, other studies such as those conducted by Cuesta et al., (2018) use ozone model data without smoothing by averaging kernels as complement to lowermost tropospheric ozone satellite data.

Some of these aspects are indicated in the RM as (lines 202–208) "This model data provides information with fine vertical resolution on the spatial distribution at these atmospheric pollutants at specific vertical levels. Therefore, they are complementary with respect to satellite retrievals, that represent the abundances of these pollutants integrated vertically within an atmospheric layer (i.e., the LMT for ozone and the total column of CO, described respectively in sections 2.1 and 2.2) and their sensitivity may vertically vary depending on the observing conditions. The combined use of model and satellite data allow accounting for the possible uncertainties or limitations of each of them, while providing a more complete description of the spatio-temporal evolution of ozone precursors.."

- Line 188: Suggestion to rephrase: "Cuesta et al. (2018) used the same data product jointly with IASI+GOME2."

Reply 18: Done (line 229–230).

- Line 242: Suggestion to rephrase: "... daily mean windspeeds lower than i) 3.2 m s–1 at 10 m and ii) lower than and 13.0 m s-1 at 500 hPa, and iii) daily total precipitation less than 1.0 mm."

**Reply 19: Done (line 282–283).**

- Lines 250 and following: "The major ozone outbreak travelling across Europe is formed by three ozone plumes originating from the Iberian Peninsula, Western Europe and North America. Three-day forward trajectories from HYSPLIT depict the pathway of these ozone plumes (Fig. 3).": In Fig. 3, foreward trajectories are shown, which originate in northern Germany, central France and the Iberian Peninsula. How are these three plumes (named "Norhternmost plume", "Middle latitude plume", and "Southermost plume") connected with the plumes originating from the Iberian Peninsula, Western Europe and North America, which are mentioned in the text? The authors should tell the reader, how they know, where the plume comes from (if not from the plots shown), and further should be consistent with the naming of the plumes.

Reply 20: Agreed and corrected. In the RM, we now describe these trajectories of the ozone plumes (now as Fig. 4) according to their forward transport and not their origin. Indeed, we named three plumes "southernmost", "middle latitude" and "northernmost plumes" based on the main transport pathway across Europe (see rectangles in Fig. 5). The three-day forward trajectories (Fig. 4) are calculated from the location where these ozone plumes are first observed over Europe. The origins of three plumes were investigated in subsection 3.1.2 (southernmost plume), 3.2.1 (middle latitude plume) and 3.2.2 (northernmost plume). For clarity, we now introduce the origin of the plumes in these sub-sections of the RM and not before.

- Line 251: Figure 3 is poorly introduced and it is not clear at this point of the paper, why this figure is needed at all.

**Reply 21: Clarified in reply 22.**

- Line 252 and following: The definition of the "plumes" is not very clear. Are these plumes connected with the trajectories shown in Fig. 3? Later on, there are references to the plume definitions, but they use figures, which have not been introduced at this point.

Reply 22: Agreed and clarified. We indeed define the location of plumes based on the three-day forward trajectories and the horizontal extent of high ozone concentration plumes depicted by IASI+GOME2 (line 308–312). This is mainly based on a clear visual distinction with respect to background concentrations in IASI+GOME2 ozone maps. The current Figure 4 (previous Figure 3) is important to track the transport partners of these ozone plumes. This is particularly useful when the boundaries between plumes are not very clearly observed since for example they are mixing into a single plume. By changing the order of the figures, this is clearer and better introduced in the RM.

- Line 256: What is considered to be a "high concentration" of ozone? Do the authors use a threshold here? What is the threshold?

Reply 23: Clarified. We often consider high ozone concentrations as roughly those above 60 ppb, although this is mainly done by visual inspection with respect to background levels of ozone in the IASI+GOME2 maps.

- Line 257: I suggest to provide the threshold for informing the population also in ppb,

since this unit has been used throughout the paper so far. Further, all of the shown measurement and simulation results are well below 90 ppb, so I would not agree to the formulation "often near 90 ppb)

Reply 24: Agreed and corrected. Indeed, the ozone concentrations in only parts of these plumes are around 80 ppb, which is near the information threshold (90 ppb for 1013 hPa and 20°C).
This is written in the RM as (lines 354–356) "The ozone concentration in part of the plumes is often around 80 ppb, which is near the ozone information threshold (whereby a 1-hour average concentration of 90 pph — for a pressure of 1013 hPa and a

a 1-hour average concentration of 90 ppb — for a pressure of 1013 hPa and a temperature of 20 °C — triggers an obligation to inform the population on possible risk; EC, 2008)."

- Figure 4: I miss a discussion, which compares the measured ozone to the TCR-2 simulated ozone. For me it looks like that there are considerable differences between panels a and b, and that it should be well motivated, why it is still useful to look into other TCR-2 species, if the simulated ozone is backed so poorly by the measurements.

Reply 25: This aspect is clarified in reply 4. We have added a comment here for clarification (lines 202–208) "This model data provides information with fine vertical resolution on the spatial distribution at these atmospheric pollutants at specific vertical levels. Therefore, they are complementary with respect to satellite retrievals, that represent the abundances of these pollutants integrated vertically within an atmospheric layer (i.e., the LMT for ozone and the total column of CO, described respectively in subsections 2.1 and 2.2) and their sensitivity may vertically vary depending on the observing conditions. The combined use of model and satellite data allow accounting for the possible uncertainties or limitations of each of them, while providing a more complete description of the spatio-temporal evolution of ozone precursors."

- Line 294: Suggestion: "As typical for Iberian summer, ..."

Reply 26: Done (line 384–385).

- Figure 5b/c: I suggest to mark the latitude/longitude (respectively), where the other cross section is located.

Reply 27: Done. We have added marks to make them easy to understand the locations visually (currently Fig. 9a–c).

- Figure 5e: This panel is not mentioned in the text. The authors should introduce it in the text or remove this panel, if it is not important for the manuscript.

Reply 28: Done. We have added the corresponding motivation for this panel (line 419–420) "This is also the case over the Iberian Peninsula on 15–17 July 2017 according to the air stagnation indicator shown in Figures 9e and 10b (navy line)." - Figure 6, caption: typo: blown -> brown

**Reply 29: Done (current Fig. 10).**

- Figure 6b: y-axis label "Percentage" is not very informative and should be replaced (e.g. by "heatwave extension/air stagnation")

**Reply 30: Done (current Fig. 10b).**

- Line 341: "... lack of sensitivity (or spatial coverage) in the satellite data (particularly over the ocean).": Is anything like such a lack of sensitivity known to be typical for the IASI+GOME2 data? Please give a reference here. Further, how much does the mismatch of observed and simulated ozone affect the following interpretation, which is only based on model data? What about mismatches between IASI+GOME2 and TCR-2 ozone above land (e.g. north-west Africa, or Turkey)?

Reply 31: Clarified. Indeed, this lack of sensitivity over ocean is well known for this kind of satellite observations (see e.g., Cuesta et al., 2013). We have added a brief explanation about the difference of sensitivity of IASI+GOME2 depending on the observing conditions and type of surface (lines 435–437). The sensitivity of IASI+GOME2 is enhanced over land with respect to that over ocean according to respectively larger and smaller thermal contrast between the near-surface air and the surface itself (Cuesta et al., 2013).

Since there is such a mismatch between the model and the satellite data, we have added further evidence for supporting the interpretation in the new Figure 11. At the beginning of the event over the Iberian Peninsula, the ozone outbreak might be affected by downward transport of ozone-rich air masses from North America and/or Western Africa located in the middle troposphere. However, omega vertical velocity over the Iberian Peninsula is not so strong (Fig. 11). On the other hand, the concentrations of ozone precursors depicted by TCR-2 are particularly high (Fig. 3). Therefore, this evidence suggests that most likely that the main source of this ozone outbreak over the Iberian Peninsula are biogenic emissions of ozone precursors. This is further explained in reply 42 for reviewer 2.

- Line 384: "In addition, this outbreak may also be affected by downward transport from middle troposphere": I don't think that this has been explained in Section 3.1.1. I also cannot see such an event in Figures 5b/c

Reply 32: We have added new figure and description about downward transport in 3.3.2. Please see in reply 42 for reviewer 2.

- Line 401 following: "It is worth noting...": Again, this sounds like there is a problem with the TCR-2 data. Why is it still useful to look into this data in the remaining of this manuscript?

Reply 33: Clarified. Please see reply 4 in "General Comments" about the motivation to use TCR-2.

- Figure 9 (and similar figures): It would be helpful to repeat the colored boxes, which mark the plumes also for other species than ozone. This would help to identify the air masses associated with the plumes.

Reply 34: Done. We have added rectangles (current Fig. 3, 5–7).

- Figure 9g/h: Please state in the caption the differences between those two panels.

Reply 35: Done. We have added the caption (current Fig. 5).

- Line 413: "However, this ozone-enriched air masses are co-located with clearly higher CO, NO2 and HCHO concentrations ...": The co-located CO plume simulated by TCR-2 is not visible in IASI measurements, as shown in Fig. 9d. In fact, this panel is not mentioned at all in the text, but it seems to be very important, since it may highlight a problem in the TCR-2 data.

Reply 36: Clarified. This CO plume is depicted by both TCR-2 and IASI data. The model simulates a CO plume uniformly distributed in the vertical from the lower troposphere to the upper troposphere. The IASI CO data corresponds to total column abundances but converted to ppb assuming that all CO molecules within the column are located in the lowermost troposphere. Therefore, both datasets depict the CO plume but TCR-2 show weaker concentrations at a given atmospheric level and higher values are seen for IASI total column data.

This is better clarified in the RM as (lines 557–559) "The simulated plume extends vertically from the lower troposphere (~930 hPa) to the upper troposphere (~240 hPa) with relatively uniform concentration (~110 ppb), thus the total column of CO is larger than the surroundings (as seen by IASI) but not so clearly at a given atmospheric level.".

- Line 442: "This ozone plume shows rather moderate concentrations ...": Please give an example or a range for "moderate concentrations".

Reply 37: Done. In the RM, this is rephrased as (line 545) "*This northernmost plume shows rather moderate concentrations* (~53 ppb, blue line in Fig. 8a)..".

- Line 470: "Significant enhancements of CO are seen ...": In Fig. 9, it was shown that TCR-2 CO may considerably disagree with IASI measurements. How is performing TCR-2 in this region in comparison to IASI?

Reply 38: Clarified. The IASI CO data also show a moderate enhancement of CO over the Balkan Peninsula as depicted by TCR-2. However, as mentioned previously, there are

other differences between TCR-2 and IASI data, a significant part of them due to the fact that this last one consists of total columns, and they are subject changes in the sensitivity of the satellite retrieval. On the other hand, the representation of biomass burning emissions in the model also likely induces uncertainties (see e.g., Huijnen et al., 2020). However, these discussions are mostly beyond the scope of this study.

- Line 539: "... show high concentrations of ozone precursors suggesting a significant impact of anthropogenic emissions ...": This statement is too general here: It depends on the specific precursor to state anthropogenic origin. I think in this case, the authors want to refer to enhanced CO levels for this plume. However, this enhanced CO in TCR-2 was shown to be not very robust compared to direct IASI measurements.

Reply 39: Clarified and corrected. We wanted to refer to the enhancement of NO2 as shown in Fig. 8e. We have clarified this in the RM (line 651–652) "Both pollution plumes show high concentrations of ozone precursors, especially NO2, suggesting a significant impact of anthropogenic emissions from Central Europe.".

- Line 542: "The satellite approach highlights significant photochemical production of ozone ...": How do the satellite measurements allow for an attribution of ozone production to photochemical production? I understood it more like the difference between IASI+GOME2 and TCR-2 suggested photochemical production?

Reply 40: Clarified and corrected. The satellite measurements show a sustained enhancement of ozone concentrations in the plume transported away from the biomass fire near the Black Sea. In absence of other sources or ozone advection, this enhancement could be linked to photochemical production. These statements are not related to TCR-2 data. On the other hand, TCR-2 data show a different evolution, which is a decrease of ozone abundance in time. This highlights the difficulty to model such situations where biomass burning emissions may affect photochemistry and ozone precursors abundances.

In the RM, this is rephrased as (lines ?) "The satellite approach depicts gradual enhancements of ozone during transport away from the Black Sea coast. In absence of other sources or ozone advection, this increase in ozone concentrations could be linked to photochemical production. On the other hand, TCR-2 simulates a temporal decrease of ozone concentrations for this plume. The differences between the model and the satellite data are probably associated...".

- Line 545: The beginning of this paragraph is rather repetitive from the beginning of this section.

Reply 41: Corrected. We have deleted this paragraph and merged with the paragraph in the beginning this section

In the RM, this is rephrased as (line 628–633) "We have presented a detailed study of the daily evolution and associated sources of precursors of a moderate ozone outbreak

transported across Europe in July 2017, by using IASI+GOME2 multispectral satellite observations and tropospheric chemical reanalysis TCR-2. The multispectral satellite approach offers the currently unique capacity of observing the ozone distribution in the lowermost troposphere (below 3 km a.s.l.), which is in fair consistency with surface observations. This regional to continental analysis cannot be made with in situ observations only since their spatial coverage is limited. This stresses the contribution of satellite-based approaches for observing ozone pollution provided their highly valuable horizontal coverage".

**Reviewer 2**

**Comments:**

**Overview**

The paper deals with the impact of the various sources of ozone precursors on the evolution of ozone concentrations during a high ozone episode over Europe in July 2017. I think that the paper presents an interesting analysis of data originating from various sources such as in-situ and satellite measurements as well as modelling. In my opinion, the manuscript is generally scientifically sound, and it deserves publication in ACP after considering the recommendations listed below.

Thank you for summary and following detailed comments.

**General Comments:**

I would suggest that, in addition to the direct photochemical ozone production at the surface and the boundary layer, which is quite properly presented and interpreted, the role and the influence of upper tropospheric ozone (generally at higher levels) to the lower tropospheric/boundary layer ozone should be more considered, especially under prevailing anticyclonic conditions, when it is reported that large scale subsidence movements might occur, especially at the edge of the anticyclones and at the interface with low pressure systems, given also the fact that the IASI+GOME2 satellite measurements are most sensitive at 2-3 km height, which is generally the free-tropospheric level. According to relatively recent publications the variability of free tropospheric ozone, especially over the Central and Eastern Mediterranean basin during summer (Kalabokas et al., 2013; Doche at al., 2014; Zanis et al., 2014; Akritidis et al., 2016; Gaudel et al., 2018) could be better understood, if the variability of synoptic meteorological conditions, affecting especially vertical ozone transport are considered. These characteristics are also detected in Central Europe during the warm period of the year by analyzing also IASI and IASI+GOME2 satellite measurements, although they seem to be more enhanced during spring months (Kalabokas et al., 2017; Kalabokas et al., 2020).

So, I would suggest examining also the IASI satellite upper tropospheric ozone and eventually the corresponding TCR-2 simulations as well as the charts of geopotential height and the omega vertical velocity at least, during the peak phase of the ozone episode (16-21 July 2017) and for the 700 and 500 hPa pressure levels. It seems that during this period, significant and extended downward movements are observed starting from the west and moving eastwards, following the evolution of episode, which

might explain better the observed phenomena.

Thank you for your thorough and helpful review of our paper. We have additionally examined the influence of the downward transport of ozone from the lower stratosphere. Please see below for specific responses to your concerns.

**Specific Comments:**

Page 16, line 345-347: I think that this factor should be more stressed by examining the daily omega vertical velocity measurements over the troposphere, as also noted above.

Reply 42: Agreed and completed. As suggested by the reviewer, we have examined the influence from the middle/upper troposphere to the lowermost troposphere and added new figures (Figs. 11 and S3). At the beginning of the period, positive omega vertical velocity (suggesting downward transport of air masses) is not very clear over the Iberian Peninsula, except for the northeastern part of this region (which is indeed closer to the interface between the high- and low-pressure systems, as suggested by the reviewer). However, the ozone enhancement depicted by IASI+GOME2 is mainly located over the southern part of the Iberian Peninsula, suggesting that its origin is likely a different one. Moreover, vertical profiles of ozone from TCR-2 show the downward transport of ozone but down to 500 hPa over the Iberian Peninsula (thus above the lowermost troposphere). Therefore, we conclude that the downward transport from the lower stratosphere and upper troposphere induced an enhanced of ozone concentrations at the middle troposphere but has less affected the lowermost atmosphere layers (below 3 km of altitude).

These aspects are detailed in the RM as (lines 443–461) "Tropospheric ozone concentrations near the Iberian Peninsula can be affected by subsidence of ozone-rich air masses from the upper to the lower troposphere, as often found during anticyclonic conditions over the Mediterranean basin during summer (e.g., Akritidis et al., 2016; Doche et al., 2014; Kalabokas et al., 2013; 2020; Zanis et al., 2014). Figure 11 describe these aspects in terms of meteorological conditions and middle/upper tropospheric ozone on 15 July. Low-pressure systems are observed over Eastern Europe while anticyclonic conditions prevail over Western Europe and the Atlantic (Fig. 11c-d). Some traces of strong subsidence are remarked over France and north of the Iberian Peninsula, between the high- and low-pressure systems, as shown by positive omega vertical velocity at 700 hPa (Fig. 11f, and also at 500 hPa, not shown). Some subsidence is also seen over the Atlantic, west of the Iberian Peninsula. Similar subsidence conditions have already been observed during regional ozone episodes over the western Mediterranean in spring and summer (Kalabokas et al., 2017; 2020). IASI+GOME2 ozone columns from 6 km to 12 km in Dobson units (DU) are relatively enhanced (~25–30 DU) over Northern and part of Central Iberian Peninsula (Fig. 11b). Relatively larger concentrations of ozone are also simulated by TCR-2 over the Iberian Peninsula at 500 hPa but only the northern part at 700 hPa (respectively ~65 ppb and  $\sim$ 80 ppb, Fig. 11c–d). A transect of vertical profiles of ozone from TCR-2 along the latitude of 41°N clearly shows downward transport of ozone-rich air masses around

5–10°W (Fig. 11e), but east of 5°E the downward transport of ozone is rather limited below 700 hPa. On the other hand, IASI+GOME2 retrievals at the lowermost troposphere (Fig. 11a) clearly depict enhanced ozone concentrations over Southern Iberian Peninsula and east of 5°E over land. Therefore, this southern part of the LMT ozone plume is probably not associated with downward transport of upper tropospheric ozone, as it is not colocalised with the region affected by subsidence (i.e. Northern Iberian Peninsula and the Atlantic west and north of the peninsula). Local production of ozone is also suggested by a moderate enhancement of ozone seen at surface level at 3°W of the transect of TCR-2 profiles (Fig. 11e)."

Page 20, lines 401-404: As noted in general comments, the plotting of omega vertical velocity charts would be helpful for the better assessment of these observations.

Reply 43: Agreed and done. We have added the omega vertical velocity in a new figure (Fig.11).

Page 22, lines 424-427: As indicated earlier, the described eastward movement of air masses seem to be also accompanied with significant downward transport throughout the troposphere, so the influence of the upper layers on ozone concentrations, have to be also investigated.

Reply 44: Agreed and done. We have examined the influence from the middle/upper troposphere to the lowermost troposphere and have found a trace of downward transport in the place where the northernmost plume is located.

These aspects are detailed in the RM as (Fig. S3 and line 543–569) "Some traces of strong subsidence are observed over Central Europe as shown by positive omega vertical velocity at 700 (Fig. S3f) and 500 hPa (not shown) on 20 July. TCR-2 ozone at 700 hPa and 500 hPa show a high concentration ozone belt over Norway, Denmark, Poland and Ukraine (Fig. S3c–d), whereas IASI+GOME2 upper tropospheric ozone columns (from 6 km to 12 km) does not show high values at those locations but north of them (Fig. S3b). Vertical profile of TCR-2 ozone at 51°N shows a downward transport of ozone-rich air masses around 15–20°E (Fig. S3e). That is the place where the northernmost plume is located (Fig. 4b). Therefore, the northernmost plume might be affected by downward transport from the middle troposphere.".

Page 25, lines 481-486: It should be added that the persistent summer northerlies, known as Etesians, in the boundary layer over the Aegean are also associated with strong large-scale subsidence in the lower troposphere occurring simultaneously, thus significantly influencing surface ozone concentrations, as it is suggested form relevant publications based on vertical and satellite measurements over the area (Kalabokas et al., 2007; Kalabokas et al., 2013; Doche at al., 2014; Zanis et al., 2014).

Reply 45: Agreed and added. We have added the description "The Etesians cause the vertical downward transport of ozone-rich air masses from the upper troposphere and the lower stratosphere as well as the horizontal transport of polluted air masses (Doche et al.,

2014; Kalabokas et al., 2007; 2013; Zanis te al., 2014)." (line 596–598).

Page 29, lines 542-544: Based also on the above remarks this discrepancy could be due to the effect of large scale tropospheric subsidence, which is very frequently observed over this region in July.

Reply 46: Convection is one of the reasons of model discrepancies as mentioned in previous studies. However, we must have to examine more carefully to conclude that the subsidence causes the discrepancies based on this study.

---

## Author Response (AR2)

**Response to Reviewers of Manuscript**

The authors have considerably improved their manuscript and many of my mentioned points have been addressed. However, some issues are still open from my point of view. Line and figure numbers refer to the tracked changes manuscript.

The authors are grateful to the reviewer for the time and thought clearly put into the review and comments regarding our paper. We have addressed all their comments below and including changes accordingly into our revised manuscript, which has led to substantial improvement in clarity and completeness. The original comments from the referees are in black and our responses in blue. Please see below our point-by-point replies to the comments of the reviewers.

**Remaining major points:**
1) I think it is crucial to connect the TCR-2 model results with measurements, as intended by the manuscript. However, this connection should be done in a way that can be easily understood by the reader. In Fig. 5 (and for O3 also 3,6,7), this is not done in a good way: IASI+GOME2 ozone from surface to 3 km is presented next to TCR-2 ozone at 850 hPa, and TCR-2 CO at 850 hPa is presented next to IASI total column CO.

Reply 1: Agreed and corrected. For presenting TCR-2 model simulations in consistency with measurements, we have modified figures 3, 5, 6 and 7 so as to always present both ozone and CO maps: (i) with satellite data and model simulations next to each other, respectively on the left- and right-hand side panels, (ii) with the same color scale ranges and units and also (iii) for the same atmospheric columns as derived from satellite (i.e. surface to 3 km of altitude (a.s.l.) partial columns in ppb for $O_3$ and total columns in molecules/cm$^2$ for CO).

2) Figures 3,5,6,7: Apples could be compared to apples, but it is not done here: From a model run, all necessary data to construct a quantity comparable to the measured quantity should be available. In particular, it is an easy task to calculate (partial) columns or average VMRs from multiple model levels. In the current state, the reader sees major differences between measured and simulated O3 and CO, but it is not clear if this is due to the different kind of quantity, or if the model simulates a different world than the world the measurements were taken in. If the latter was true, all following conclusions drawn on TCR-2 results would be highly questionable. So it is very important to show that the model agrees to the measurements first.

Reply 2: Agreed and done. As described in reply 1, we have followed the reviewer's suggestion. The revised manuscript presents satellite and TCR-2 simulation maps of ozone and carbon monoxide for the same atmospheric columns, units and scales.

3) Color bars of TCR-2 CO and IASI CO have different maximum VMRs, which is very misleading, once the quantities are in a comparable shape.

Reply 3: Agreed and corrected. The revised manuscript uses the same color bars for IASI and TCR-2 CO (fig. 5).

4) Usage of TCR-2 in this manuscript: I do not agree with the answer of the authors that the usage of TCR-2 is only "complementary" and does not need to match the IASI+GOME2 observations. I think that it is a strength of this study that observations and simulations are analyzed together. Pure model studies have the weakness that they potentially could describe a fictional world, so it is much better to also include measurements, as it is done here. But once

measurements are used, it needs to be assured that measurements and models are telling the same story, or explain the differences.

Reply 4: Agreed and done as requested. As recommended, we have calculated from TCR-2 the lowermost tropospheric ozone (surface–3 km) partial columns and total columns of CO to clearly show the differences with IASI+GOME2 ozone and IASI CO, in adjacent panels with respect to satellite data. In some cases, TCR-2 simulates similar features than the observations, in orders some differences are seen mostly in absolute terms. These last ones are mostly linked to uncertainties in the model and differences in the sensitivity of the satellite retrieval. Please find below explanations of these differences provided in the revised manuscript.

Lines 431–437: "*However, those ozone plumes transported to the Iberian Peninsula are only partially depicted by IASI+GOME2 observations. These differences could be associated with misrepresentation of these ozone plumes in the simulations or lack of sensitivity (or spatial coverage) in the satellite data (particularly over the ocean). This last aspect is related to a reduction of the sensitivity of IASI+GOME2 over the ocean, due to smaller thermal contrasts than over land (Cuesta et al., 2013). The 7-day back-trajectories confirm that the air masses come from the west over the North Atlantic and offshore transport from Western Africa in the middle troposphere (Fig. 3f). It suggests that the link between those ozone plumes and emissions in North America is unclear.*"

Lines 509–517: "*The tropospheric reanalysis also depicts a denser ozone plume over Northern Spain, Western France and the nearby Atlantic. This last one is not depicted by IASI+GOME2 at the LMT nor by in situ measurements at the surface. Ozone concentrations simulated at surface level (not shown) are rather moderate at this location, but higher over Eastern France, thus in better agreement with satellite data. At the upper troposphere (between 6 and 12 km of altitude, not shown), IASI+GOME2 does depict an ozone plume over Western France and located slightly north of the one simulated by TCR-2. This suggests uncertainties in the vertical and horizontal location of this lofted ozone plume in the model. A limitation of TCR-2 may come from the fact that the satellite ozone observations assimilated in TCR-2 are derived from TES measurements with a coarse horizontal resolution (only nadir pointing) and most sensitivity to ozone at the free troposphere at lowest.*",

Lines 555–557: "*In this location, TCR-2 reanalyses depict rather weak increases in background amounts of CO between the east coast of England and Eastern Europe (Fig. 5d). This overestimation can be attributed to errors in the surface emission, chemical productions and losses, long-range transport from North America. On the other hand, the amount of CO over the Black Sea and the Eastern Mediterranean is comparable.*"

Lines 617–625: "*On the other hand, TCR-2 simulates a temporal decrease of ozone concentrations for this plume. The differences between the model and the satellite data are probably associated with the representation of biomass burning emissions and its impact on the simulation of ozone production. Simulations of these processes and their quantification are currently challenging, as shown by the large discrepancies between different state-of-the-art global tropospheric ozone reanalysis products (e.g., Huijnen et al., 2020). Near-surface ozone concentrations are weakly and mainly indirectly constrained by the satellite observations used for assimilation in reanalysis such as TCR-2. In addition, only very few ground-based stations monitor air pollutants over Eastern Europe, as their geographical coverage is far from homogenous across the globe.*

*Particularly in those locations, satellite observations of lowermost tropospheric ozone as those derived by IASI+GOME2 are particularly appreciated for filling the gap of ground-based observations.*"

**Remaining minor points:**
1) Figure 5: The sorting of the panels is inconsistent, since model data is on the right side in the first row, and on the left side in the second row.

Reply 5: Agreed and done. Sorting of the panels of Fig. 5 has been modified as requested (fig. 5c–d).

2) IASI CO total column is presented in "ppb". Does that mean that the total column is again divided by the total air column to get an average VMR?

Reply 6: Clarified and modified. For showing the original units of the satellite retrieval and compare it with model data, we have changed the unit of total columns of CO to "molec. cm$^{-2}$" (fig. 5c).

3) Regarding "Reply 11": I still do not understand, which data point of the gridded satellite data set is used for comparison with the in situ data. The nearest grid point? An interpolation of the four nearest data points?

Reply 7: Clarified. The statistics considers the nearest point of IASI+GOME2 1x1° grided data.

> This is clarified in the RM as (line 135–136) "*Figure 1 shows the Pearson correlation coefficients (R) between daily in situ surface observations at each individual station (squares or circles) and the nearest grid point of IASI+GOME2 retrievals.*"

4) Regarding "Reply 12": It should be mentioned, which kind of significance test was used, if such quantities as P-values are discussed.

Reply 8: Clarified.

> This is clarified in the RM as (line 138) "*(with P-values of the Student's t test < 0.05)*"

5) Regarding "Reply 17": See major points 2 & 4

Reply 9: Please see reply 2 and 4.

6) Regarding "Reply 23": In my opinion, language is important here: If the authors want to say that the VMR is higher than the background, I would suggest to use formulations like "relatively high concentration", or "higher than background concentration". Otherwise: If there was a certain threshold used to identify the air mass, the threshold should be mentioned in the manuscript.

Reply 10: Agreed and clarified. We did not set a specific threshold to define ozone plumes in this study, therefore we have used the word "relatively high".

7) Regarding "Reply 27": It is nice that the authors now marked the altitude range, which was shown in panel a. However, I rather meant to mark the horizontal overlap of panels b and c, instead. Still, this is not very important in my view and was only a suggestion to guide the reader's eyes.

Reply 11: Done. We have marked the positions of intersection in the Fig. 9b-c.